# Plug-and-Fold:
# Weight-Preserving Structured Compression for Large Language Models

## Abstract

Large Language Models (LLMs) have achieved remarkable performance across a wide range of tasks, but their growing size poses significant challenges for deployment and efficiency. Among existing model compression methods, structured pruning has emerged as a popular approach for reducing model size. However, these methods remove structural components such as layers, heads, or channels, which can disrupt pre-trained weights and lead to fragile recovery fine-tuning process. In this work, we propose Plug-and-Fold (PnF), a weight-preserving yet structurally effective compression method. Rather than directly modifying or factorizing the pretrained weights, PnF introduces lightweight, learnable adapter modules into the projection layers of attention and feed-forward networks. These adapters are trained while keeping the original weights frozen, and are later folded into the base weights via simple matrix multiplications. This process yields a compressed model that preserves the original architecture and can be deployed with a standard Transformer inference stack, without custom kernels or additional runtime components. We evaluate PnF across a variety of benchmarks and model scales, demonstrating consistent improvements over recent state-of-the-art structured compression baselines. Our results highlight that preserving the integrity of pretrained weights not only simplifies the compression pipeline, but also improves generalization and performance recovery in compressed LLMs.

## 1 Introduction

Large language models (LLMs) based on the Transformer (Vaswani et al., 2017) have achieved remarkable progress across various domains, including natural language processing (Zhao et al., 2023; Jiang et al., 2024a; Radford et al., 2018), code generation (Jiang et al., 2024b), computer vision (Liu et al., 2023a; Hamadi, 2023), and scientific applications (Zhang et al., 2025; Lin et al., 2023). This progress is attributable to two factors: (1) scaling model size to billions to trillions of parameters (Team et al., 2024; Islam & Moushi, 2025; Team et al., 2025; Zhang & Sennrich, 2019) and (2) pre-training on massive, diverse corpora (Langlais et al., 2025; Liu et al., 2024). Together, these endow LLMs with deep language understanding and ability to generate high-quality code, text, and multi-modal contents.

Despite these successes, their massive parameter sizes pose critical challenges: they require large storage, memory footprints, increase inference latency, and substantial computation for training and deployment, especially in resource-constrained settings. To address these practical limitations, a substantial body of research has focused on model compression techniques that shrink the footprint while preserving performance. These methods can be grouped into three principal categories: (1) knowledge distillation, which transfers capabilities from a large teacher to a smaller student (Hinton, 2014; Ojha et al., 2023; Agarwal et al., 2023; Bing et al., 2025; Cui et al., 2025); (2) quantization, which lowers numerical precision to save memory and accelerate inference (Liu et al., 2023b; Li et al., 2024b; Shang et al., 2023; Hu et al., 2025; An et al., 2025); and (3) pruning, a structured approach that removes redundant channels, heads, or layers (Voita et al., 2019; Gao et al., 2024b; Ma et al., 2023; Ashkboos et al., 2024; Men et al., 2024; Mugnaini et al., 2025; Yang et al., 2024).

Pruning gained a lot of attention since it leverages the pre-trained weights of the original model and typically does not require to training a new network from the ground up. Moreover, once the unnecessary components have been eliminated, the resulting model can be further compressed through quantization, yielding additional reductions in memory consumption and inference latency.

{In the context of LLMs, structured compression has primarily focused on pruning-based methods, such as deleting channels from the projection weights in attention and feed-forward networks (Ashkboos et al., 2024; Gao et al., 2024b; Ma et al., 2023), removing heads in the multi-head attention (Voita et al., 2019; Mugnaini et al., 2025), and pruning whole Transformer layers (Yang et al., 2024; Men et al., 2024). The selection of components to prune is guided by metrics that estimate the impact of removal, such as the magnitude of weights and activations (Sun et al.), cosine similarity (Men et al., 2024), or the L2-norm (Ashkboos et al., 2024). Other approaches adopt learning-based structural compaction schemes, where auxiliary matrices (e.g., compactor or mask matrices) are inserted around backbone weights and jointly optimized with the original weights; after training, rows, columns, or channels of these learned structures are pruned and folded back into compressed weights (Wu et al., 2024; Hu et al., 2024). Although these approaches leave the overall Transformer architecture intact, their joint optimization scheme still perturb parameters that were carefully tuned during large-scale pretraining, often leading to non-trivial performance loss.

Consequently, many approaches incorporate a recovery fine-tuning (RFT) stage to restore accuracy, often employing the lightweight adapter such as LoRA (Voita et al., 2019; Gao et al., 2024b; Ma et al., 2023; Ashkboos et al., 2024; Men et al., 2024; Mugnaini et al., 2025; Yang et al., 2024). However, the recovery process can be fragile: even extensive RFT often fails to fully restore the performance of precisely optimized foundation models.

To overcome these limitations, we propose a **weight-preserving** structured compression that retain the integrity of pretrained weight while still achieving substantial efficiency gains. Our method, Plug-and-Fold (PnF), inserts lightweight, learnable adapter modules into the original projection matrices of the attention and feed-forward sub-layers rather than removing heads, channels, or layers.

In contrast to prior pruning and low-rank approaches that directly modify original pretrained weights, PnF freezes all pretrained weights throughout compression and trains only lightweight adapter modules attached to the original model, thereby formulating model compression as a PEFT-style training problem that preserves the expressivity and knowledge encoded in the original model. After training, the learned adapters are folded into a single dense matrix via simple matrix multiplications; PnF therefore uses PEFT-style adapters as a tool for structured compression and deploys a compressed model that is structurally identical to the original, rather than directly editing or factorizing the backbone.

Because no architectural modification is introduced and no extra operations are required during inference, PnF can be integrated seamlessly into existing serving frameworks and hardware accelerators.

We evaluate PnF with extensive experiments covering a broad spectrum of model sizes and compression rates. To validate its effectiveness, PnF is benchmarked against the latest state-of-the-art structured-compression baselines on a diverse set of tasks that demand varied domain knowledge and comprehensive capabilities. Across all settings, PnF consistently surpasses existing methods, delivering notable gains in downstream performance. These results show that preserving the integrity of pretrained weights not only yields a simpler and more scalable compression pipeline, but also enhances the recovery of accuracy and the generalization ability of the compressed models.

The main contributions of our paper are summarized as follows:

- We propose Plug-and-Fold (PnF), a novel weight preserving structured compression method that inserts lightweight, learnable adapter modules into the original projection layers without modifying the model architecture.

- After training, the adapters are folded into the base weights via simple matrix multiplications, resulting in a compressed model that is structurally identical to the original model and reduces runtime effectively.

- Extensive experiments demonstrate that PnF outperforms recent state-of-the-art structured-compression baselines across a wide range of model scales and benchmark tasks, confirming its effectiveness and scalability.

## 2 BACKGROUND

### 2.1 DECODER-BASED TRANSFORMER ARCHITECTURE

Large Language Models (LLMs) primarily leverage a decoder-based Transformer architecture composed of stacked decoder blocks. These blocks consist of two core components: the Multi-Head Self-Attention (MHSA) mechanism and the Feed Forward Network (FFN). These components form the core layers of decoder blocks, enabling sequential data processing and contextual understanding.

#### 2.1.1 MULTI-HEAD SELF-ATTENTION (MHSA)

The MHSA mechanism enables the model to dynamically weight and aggregate contextual information from different positions in the input sequence by utilizing attention heads. Formally, let the $l$-th decoder block takes input hidden state $X^{(l-1)} \in \mathbb{R}^{n \times d_{\text{embed}}}$, where $n$ and $d_{\text{embed}}$ is the length and the dimension of the input, respectively. For the $i$-th attention head, $i \in \{1, \cdots, n_h\}$, the MHSA mechanism computes the query vectors $Q_i^{(l)} \in \mathbb{R}^{n \times d_{\text{head}}}$, key vectors $K_i^{(l)} \in \mathbb{R}^{n \times d_{\text{head}}}$, and value vectors $V_i^{(l)} \in \mathbb{R}^{n \times d_{\text{head}}}$ as follows:

$$Q_i^{(l)} = X^{(l-1)}W_{Q_i^{(l)}}, \quad K_i^{(l)} = X^{(l-1)}W_{K_i^{(l)}}, \quad V_i^{(l)} = X^{(l-1)}W_{V_i^{(l)}}, \tag{1}$$

where $W_{Q_i^{(l)}}, W_{K_i^{(l)}}, W_{V_i^{(l)}} \in \mathbb{R}^{d_{\text{embed}} \times d_{\text{head}}}$ are the learned weight parameters for query, key, and value projections, and $d_{\text{head}}$ is the dimension of the head (often $d_{\text{head}} = \frac{d_{\text{embed}}}{n_h}$). Then, the self-attention operation is applied to each triple $(Q_i^{(l)}, K_i^{(l)}, V_i^{(l)})$ and computes the attention output of the $i$-th head $Z_i^{(l)}$ as follows:

$$Z_i^{(l)} = \text{Attention}(Q_i^{(l)}, K_i^{(l)}, V_i^{(l)}) = \text{Softmax}\left(\frac{Q_i^{(l)}\left(K_i^{(l)}\right)^\top}{\sqrt{d_k}}\right)V_i^{(l)}, \tag{2}$$

where $\sqrt{d_k}$ is a scaling factor applied to ensure numerical stability. To represent comprehensive contextual information, these outputs from individual heads are concatenated and transformed as follows:

$$Z^{(l)} = \text{Concat}(Z_1^{(l)}, \cdots, Z_h^{(l)})W_{O^{(l)}} \in \mathbb{R}^{n \times d_{\text{embed}}}, \tag{3}$$

where $\text{Concat}(\cdot)$ is the concatenation operation and $W_{O^{(l)}} \in \mathbb{R}^{(h d_{\text{head}}) \times d_{\text{embed}}}$ is learned weight parameters for output.

#### 2.1.2 FEED-FORWARD NETWORK (FFN)

Following the MHSA mechanism, the output is passed through a Feed Forward Network (FFN) to enhance the model's capacity to process through non-linear transformations and increased number of parameters. The FFN is often applies linear transformations separated by a nonlinear activation function $\sigma(\cdot)$ (e.g., SiLU(Elfwing et al., 2018)). For example, SwiGLU (Shazeer, 2020) module is defined as follows:

$$\text{SwiGLU}(Z^{(l)}) = \left(\sigma(Z^{(l)}W_{\text{gate}^{(l)}}) \odot Z^{(l)}W_{\text{up}^{(l)}}\right)W_{\text{down}^{(l)}} \tag{4}$$

where $\sigma$ is the Swish activation function (Ramachandran et al., 2018) , and $W_{\text{gate}^{(l)}}, W_{\text{up}^{(l)}} \in \mathbb{R}^{d_{\text{embed}} \times d_{\text{inter}}}$, and $W_{\text{up}^{(l)}} \in \mathbb{R}^{d_{\text{inter}} \times d_{\text{embed}}}$ are learnable parameters with the intermediate dimension $d_{\text{inter}}$.

## 3 METHOD

In this section, we present Plug-and-Fold (PnF) compression, a straightforward yet effective compression method for large language models, whose complete workflow is illustrated in Figure 1[1].

---

[1] Snowflake and Fire icons created by Freepik – Flaticon

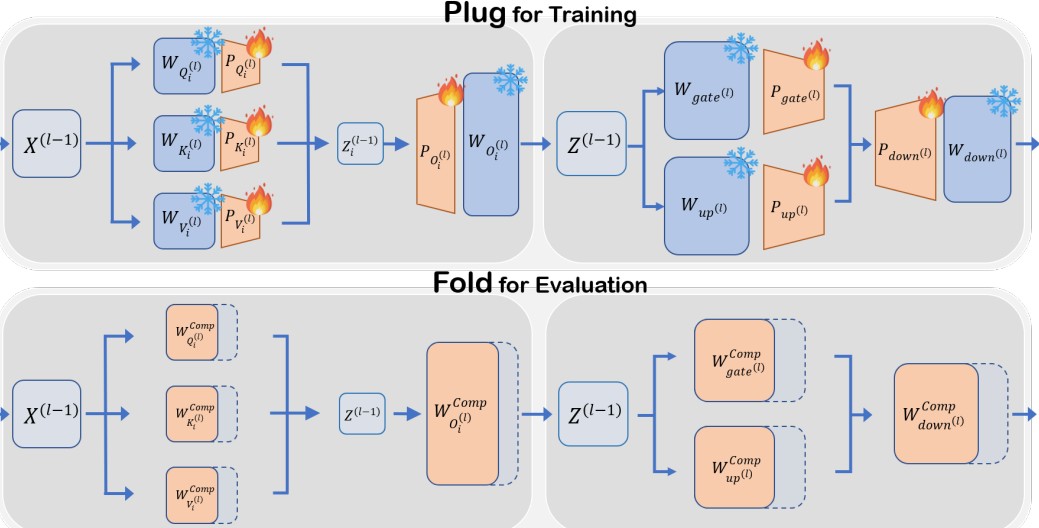

Figure 1: **Visualization of Plug-and-Fold framework**. The top half illustrates the training phase: lightweight PnF adapters are *plugged* into the pretrained linear layers and project to a reduced-dimensional space; the backbone weights remain frozen (shown as snowflakes), while the adapters are the only trainable components (shown as fire), enabling them to fully leverage the already-optimized structure. The bottom half shows the evaluation phase: after training, each adapter is *folded* back into its corresponding weight matrix via a simple matrix multiplication, yielding a compressed model that preserves the original architecture, interface, and performance.

The main objective of this method is to preserve the original projection weight during training while reducing their dimensionality, yeidling a compact model that maintains the original signal.

Section 3.1 introduce the PnF adapter, a foldable compression module plugged into the original projection weights and trained to induce low-dimensional projection while preserving the original signal. Section 3.2 presents training schemes used to train these adapters effectively. Finally, Section 3.3 describes how the trained PnF adapters are folded into low-dimensional projection weights, producing a compact model that is computationally efficient while preserving performance suitable for deployment.

### 3.1 PLUG-AND-FOLD (PNF) COMPRESSION

#### 3.1.1 PLUG-AND-FOLD (PNF) ADAPTER

In order to preserve the original signal while training, Plug-and-Fold adapters are plugged into the pre-trained model. Given a pre-trained linear weight $W \in \mathbb{R}^{m \times n}$, we define the PnF adapter as a linear projection:

$$P \in \mathbb{R}^{n \times r}, \tag{5}$$

where $r < n$. The adapter is applied to $W$ and subsequently trained to recover the performance of the original model. Formally, our aim is to find an adapter $P$ that satisfies:

$$\mathcal{P}(W) \approx \mathcal{P}(WP), \tag{6}$$

where $\mathcal{P}(\cdot)$ denotes the performance measures on various tasks induced by the corresponding weight. Consequently, projecting the weights through the trained adapter $P$ that satisfies Eq. (6) yields output representations in the reduced-dimensional space ($r$-dimension), while preserving a quality comparable to that of the full-size model. i.e., this projection yields compact representations that preserve the fidelity of the original weight matrix, allowing highly efficient deployment across a broad range of downstream tasks.

#### 3.1.2 PNF ADAPTER FOR MHSA

We now explain how PnF adapter is integrated into the MHSA layer of an LLM. Let the projection weights for queries, keys, values and the output at layer $l$ be $W_{Q_i^{(l)}}, W_{K_i^{(l)}}, W_{V_i^{(l)}} \in \mathbb{R}^{d_{\text{embed}} \times d_{\text{head}}}$,

and $W_{O^{(l)}} \in \mathbb{R}^{(n_h d_{\text{head}}) \times d_{\text{embed}}}$, where $n_h$ is the number of attention heads. For each of these matrices, we plug in a corresponding PnF adapter with dimension $r_{\text{head}} < d_{\text{head}}$:

$$P_{Q_i^{(l)}}, \ P_{K_i^{(l)}}, \ P_{V_i^{(l)}} \in \mathbb{R}^{d_{\text{head}} \times r_{\text{head}}^{(l)}}, \ \text{and} \ P_{O^{(l)}} \in \mathbb{R}^{(n_h r_{\text{head}}^{(l)}) \times (n_h d_{\text{head}})} \tag{7}$$

These adapters, multiplied with the original weights, produce lower-dimensional projections:

$$
\begin{aligned}
W_{Q_i^{(l)}} P_{Q_i^{(l)}} &\in \mathbb{R}^{d_{\text{embed}} \times r_{\text{head}}^{(l)}} \\
W_{K_i^{(l)}} P_{K_i^{(l)}} &\in \mathbb{R}^{d_{\text{embed}} \times r_{\text{head}}^{(l)}} \\
W_{V_i^{(l)}} P_{V_i^{(l)}} &\in \mathbb{R}^{d_{\text{embed}} \times r_{\text{head}}^{(l)}} \\
P_{O^{(l)}} W_{O^{(l)}} &\in \mathbb{R}^{(n_h r_{\text{head}}^{(l)}) \times d_{\text{embed}}}
\end{aligned}
\tag{8}
$$

Thus, each attention projection incorporates a learnable low-rank adapter. After training, folding the adapter into the original weight via matrix multiplication gives substantial reduction in both memory usage and computational overhead, while maintaining output quality of the uncompressed model.

### 3.1.3 PnF adapter for Feed Forward Network

Next, we present the applicaiton of PnF adapters to the FFN. Let the gate, up-projection, and down-projection at layer $l$ be $W_{\text{gate}^{(l)}}, W_{\text{up}^{(l)}} \in \mathbb{R}^{d_{\text{embed}} \times d_{\text{inter}}}$, and $W_{\text{down}^{(l)}} \in \mathbb{R}^{d_{\text{inter}} \times d_{\text{embed}}}$, respectively. For these matrices, we introduce the corresponding PnF adapters:

$$P_{\text{gate}^{(l)}}, P_{\text{up}^{(l)}} \in \mathbb{R}^{d_{\text{inter}} \times r_{\text{inter}}^{(l)}}, \ \text{and} \ P_{\text{down}^{(l)}} \in \mathbb{R}^{r_{\text{inter}}^{(l)} \times d_{\text{inter}}} \tag{9}$$

where $r_{\text{inter}}^{(l)} < d_{\text{inter}}^{(l)}$. Multiplying these adapter with the original weights yields the compressed projections:

$$
\begin{aligned}
W_{\text{gate}^{(l)}} P_{\text{gate}^{(l)}} &\in \mathbb{R}^{d_{\text{embed}} \times r_{\text{inter}}^{(l)}} \\
W_{\text{up}^{(l)}} P_{\text{up}^{(l)}} &\in \mathbb{R}^{d_{\text{embed}} \times r_{\text{inter}}^{(l)}} \\
P_{\text{down}^{(l)}} W_{\text{down}^{(l)}} &\in \mathbb{R}^{r_{\text{inter}}^{(l)} \times d_{\text{embed}}}
\end{aligned}
\tag{10}
$$

Therefore, similar to that of the attention mechanism with PnF adapters above, each FFN layer is equipped with a learnable low-rank adapter. Because the feed-forward network (FFN) comprises the majority of a transformer's parameters, folding the adapters into the original weights provides substantial savings in both memory and computation.

### 3.2 Training Pipeline for PnF adapter

To obtain PnF adapters with high fidelity, we propose a three-stage training pipeline: (i) **Compression Planning** that determines the per-layer degree of dimensionality reduction, (ii) **Group-wise Sequential Training** that stabilizes optimization by sequentially training a small, isolated set of adapters, and (iii) **KL-divergence Distillation Loss** that aligns the compressed model's output distribution with the original model's distribution.

**Stage 1: Compression Planning** Based on desired compression ratio (e.g., 20%), we first determine the degree of reduction of dimensionality (i.e., $r_{\text{head}}^{(l)}$ and $r_{\text{inter}}^{(l)}$) for each layer $l$. While the allocation of reductions can be flexible, we recommend a pyramidal schedule where deeper layers (closer to the language modeling head) are compressed more aggressively, and earlier layers receive milder reductions. Prior work on layer pruning Men et al. (2024); Gromov et al. (2024) shows that later (upper) layers can often be removed with little impact on downstream performance, indicating that they contribute less to the model's expressivity. Based on this finding, we allocate a larger portion of the compression budget to the top of the model.

Because the reduction ratio can be explicitly set, the approach is highly flexible and can be tailored to meet a user's requirements. Our empirical studies reveal that applying a higher compression rate to the FFN yields considerably better results than compressing the MHSA modules, and a concrete example of this planning is provided in the Appendix B.

**Stage 2: Group-wise Sequential Training**  Plugging all adapters at once might perturb the original model's signal at the beginning of training, inducing covariate shift and misleading gradients. Alternatively, training a single adapter at a time preserves this signal but is prohibitively slow. To address this issue, we introduce Group-wise Sequential Training. This training scheme trains small groups of adapters in turn, retaining most of the signal preservation benefits while substantially reducing training time and stabilizing convergence, which is further discussed in Section 4.3.1. Formally, we first partition the $L$ transformer layers into disjoint groups of size $N$, starting from the top of the model (output side) and moving downward. The $k$-th group is defined as:

$$\mathcal{G}_k = \{L - kN + 1, \cdots, L - (k-1)N\}, \quad k = 1, 2, \cdots, n_g, \tag{11}$$

where $n_g = \lfloor L/N \rfloor$ is the number of groups. Given the compression plan that specifies per-layer reductions (i.e., $r_{\text{head}}^{(l)}$ and $r_{\text{inter}}^{(l)}$), we first identify which group contain layers slated for compression. Then training proceeds sequentially from $\mathcal{G}_1$ towards $\mathcal{G}_{n_g}$.

At step $k$, if $\mathcal{G}_k$ includes layers selected by the compression plan, we insert adapters only into those layers and train them, while keeping the adapters trained in previous groups ($\mathcal{G}1, \ldots, \mathcal{G}_{k-1}$) frozen. During this phase, only the parameters of current group are updated; all previous groups remain frozen with their trained adapters, while remaining groups ($\mathcal{G}_{k+1}, \cdots, \mathcal{G}_{n_g}$) remain frozen without adapters (i.e., in their original state).

An instance of group-wise sequential training is illustrated in Figure 2, given $L = 36$ and $N = 4$, the compression plan targeting layers 13 - 36 covers six groups ($\mathcal{G}_1, \cdots, \mathcal{G}_6$). We train these six groups sequentially from the output side toward the input (i.e., $\mathcal{G}_1 \rightarrow \cdots \rightarrow \mathcal{G}_6$) while the lower 12 layers remain uncompressed. By activating one small group per step and keeping the remaining group fixed, this approach preserves the backbone signal and improves optimization stability.

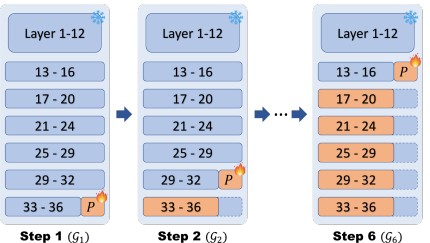

**Stage 3: KL-divergence Distillation Loss**  During the group-wise sequential training for the adapters, we adopt a Kullback-Leibler (KL) divergence loss. Specifically, the logits of the PnF-plugged model are aligned with those of the frozen backbone model by minimizing:

Figure 2: Visualization of group-wise sequential training. Training proceeds group by group, beginning with the output side. At any step, only the current group $\mathcal{G}_i$ is updated while all other groups stay frozen, which preserves the backbone signal and enhances optimization stability.

$$\mathcal{L}_{\text{KL}} = \text{KL}(p_W \| p_{WP}) \tag{12}$$

where $p_W$ and $p_{WP}$ denote the predictive distribution of the backbone and the PnF-plugged models, respectively.

We adopt a KL-divergence distillation loss for two reasons. First, the goal of compression is to produce a smaller model that reproduces the original model's behavior. The KL-divergence can achieve this by aligning the predictive distribution of student (PnF-plugged model) with the teacher (original model). Second, recent studies (Bercovich et al., 2024; Muralidharan et al., 2024; Li et al., 2024a) report that KL-based distillation often outperforms cross-entropy, yielding better downstream performance.

### 3.3 DEPLOYMENT FOR INFERENCE

After the adapters are fully trained leveraging unhindered pre-trained weights, they can be seamlessly integrated into the backbone model. In MHSA, for example, each adapter is folded into its corresponding pre-trained weight matrix via matrix multiplication:

$$
\begin{aligned}
W_{Q_i^{(l)}} P_{Q_i^{(l)}} &\rightarrow W_{Q_i^{(l)}}^{\text{Comp}} \\
W_{K_i^{(l)}} P_{K_i^{(l)}} &\rightarrow W_{K_i^{(l)}}^{\text{Comp}} \\
W_{V_i^{(l)}} P_{V_i^{(l)}} &\rightarrow W_{V_i^{(l)}}^{\text{Comp}} \\
P_{O^{(l)}} W_{O^{(l)}} &\rightarrow W_{O^{(l)}}^{\text{Comp}}
\end{aligned}
\tag{13}
$$

A similar folding procedure applies to FFN, where each adapter is integrated into its corresponding weight matrix:

$$W_{\text{gate}(l)} P_{\text{gate}(l)} \rightarrow W_{\text{gate}(l)}^{\text{Comp}}$$
$$W_{\text{up}(l)} P_{\text{up}(l)} \rightarrow W_{\text{up}(l)}^{\text{Comp}} \quad (14)$$
$$P_{\text{down}(l)} W_{\text{down}(l)} \rightarrow W_{\text{down}(l)}^{\text{Comp}}$$

The resulting weights directly replace the original model, reducing parameter counts and computational costs while preserving the model's architectural structure and inference pipeline. This fold-in operation has two key benefits. First, deployment is simple: the trained PnF adapters are folded into the original weights via plain matrix multiplications– no auxiliary metrics, graph edits, or specialized operators. Second, it ensures that the deployed model remains identical structure and interface to the original model, which facilitates compatibility with existing serving frameworks and hardware accelerators.

In practice, layer-wise non-uniform width patterns used in PnF have been empirically shown to be deployment-friendly. Both elastic Transformer designs, such as MatFormer-style models deployed in Gemma 3n Devvrit et al. (2024); Google DeepMind (2025), as well as adaptive pruning and compression methods that allocate capacity under a global budget Ban et al. (2025); Yang et al. effectively support irregular per-layer widths on standard dense general matrix to matrix multiplication (GEMM) inference pipelines and existing serving frameworks.

We concisely summarize the PnF compression pipeline: compression planning, group-wise adapter training, and the final folding step in Appendix A

## 4 EXPERIMENTS

In this section, we first evaluate the PnF Compression method against several widely-used compression methods across different compression rates and original model sizes, demonstrating its effectiveness (Section 4.2). We then examine the impact of our weight-preserving mechanism and training strategies through an ablation study (Section 4.3).

### 4.1 EXPERIMENTAL SETUP

All experiments were conducted to systematically compare the effectiveness of various large language model (LLM) compression techniques across a suite of widely-used benchmark tasks. We evaluated each method Slice-GPT (Ashkboos et al., 2024), LaCo (Yang et al., 2024), ShortGPT (Men et al., 2024), LLM-Streamline (Chen et al., 2025), and our proposed method in three target compression rates (approximately 20%, 30%, and 40%) relative to the original model size. The baselines consist of the uncompressed models: Qwen3-4B-Base, Qwen3-8B-Base, OPT 2.7B, OPT 6.7B, LLaMA-3.2-3B, and LLaMA-3.1-8B. We additionally report comparisons on LLaMA-2-7B at a 20% compression ratio against latest structured compression and quantization/distillation baselines—SVD-LLM Wang et al. (2024), Bit-Distiller Du et al. (2024), LLM-Pruner Ma et al. (2023), and DISP-LLM Gao et al. (2024b) in Appendices F and G. Moreover, we report task performance and average per-token inference latency for the original backbone, SliceGPT, and PnF across various compression ratios in Appendix H to assess the impact of compression on generation speed.

The evaluation benchmarks include: PIQA (physical commonsense reasoning), HellaSwag (commonsense inference), WinoGrande (pronoun resolution), CSQA (commonsense QA), ARC-e/ARC-c (science questions), OpenBookQA, BoolQ (boolean QA), Social IQA (multiple-choice), MMLU (multi-task language understanding), and Lambda OpenAI (factual QA). Each model's performance is measured using task-specific accuracy, or accuracy norm if available, reported per dataset. For each compression approach and setting, we tabulate the compression rate (CR) and all benchmark scores, along with the average performance (AVG) across tasks and relative performance rate (RP).

For a fair comparison, all compressed models underwent a performance recovery phase following the respective compression procedure. Specifically, our approach utilizes adapter training for post-compression recovery; the Streamline baseline employs light layer training; and other methods adopt LoRA (Hu et al., 2022) training as their recovery protocol. All recovery procedures leveraged the SlimPajama dataset (Soboleva et al., 2023), sampling 600,000 training instances, each with a

Table 1: Performance of the various compression methods on Qwen-3-8B-Base. The pretrained backbone model and its compressed variants are evaluated across multiple benchmarks at several compression rates. The best and second-best results at each compression rate are highlighted with **boldface** and underline, respectively.

| Method | CR | PIQA | HS | WG | CSQA | ARC-e | ARC-c | OBQA | boolq | SIQA | mmlu | ld | Avg | RP |
|---|---|---|---|---|---|---|---|---|---|---|---|---|---|---|
| Baseline | 0% (8.19B) | 0.793 | 0.786 | 0.724 | 0.860 | 0.801 | 0.573 | 0.410 | 0.830 | 0.547 | 0.747 | 0.709 | 0.707 | 1.000 |
| Slice GPT | 20% (6.52B) | 0.716 | 0.617 | 0.665 | 0.195 | 0.644 | 0.401 | 0.376 | 0.749 | 0.418 | 0.247 | 0.571 | 0.509 | 0.720 |
|  | 30% (5.71B) | 0.667 | 0.544 | 0.624 | 0.199 | 0.511 | 0.317 | 0.362 | 0.601 | 0.404 | 0.231 | 0.505 | 0.451 | 0.638 |
|  | 40% (4.91B) | 0.618 | 0.447 | 0.586 | 0.194 | 0.405 | 0.263 | 0.332 | 0.523 | 0.392 | 0.230 | 0.422 | 0.401 | 0.567 |
| LaCo | 20% (6.65B) | 0.733 | 0.645 | 0.658 | 0.627 | 0.665 | 0.422 | 0.382 | 0.673 | 0.453 | 0.560 | 0.587 | 0.582 | 0.824 |
|  | 30% (5.88B) | 0.687 | 0.524 | 0.589 | 0.405 | 0.561 | 0.337 | 0.320 | 0.722 | 0.425 | 0.362 | 0.522 | 0.496 | 0.701 |
|  | 40% (5.10B) | 0.614 | 0.398 | 0.554 | 0.205 | 0.423 | 0.277 | 0.292 | 0.501 | 0.387 | 0.242 | 0.305 | 0.382 | 0.540 |
| LLM-Streamline | 20% (6.65B) | 0.757 | 0.612 | 0.559 | 0.211 | 0.647 | 0.375 | 0.400 | 0.618 | 0.441 | 0.255 | 0.508 | 0.489 | 0.692 |
|  | 30% (5.88B) | 0.717 | 0.501 | 0.534 | 0.192 | 0.524 | 0.303 | 0.348 | 0.617 | 0.393 | 0.229 | 0.358 | 0.429 | 0.606 |
|  | 40% (5.10B) | 0.589 | 0.362 | 0.571 | 0.196 | 0.356 | 0.264 | 0.286 | 0.430 | 0.376 | 0.230 | 0.017 | 0.334 | 0.473 |
| Short GPT | 20% (6.65B) | 0.632 | 0.362 | 0.513 | 0.195 | 0.439 | 0.261 | 0.300 | 0.553 | 0.368 | 0.247 | 0.070 | 0.358 | 0.506 |
|  | 30% (5.88B) | 0.608 | 0.326 | 0.507 | 0.187 | 0.416 | 0.238 | 0.286 | 0.462 | 0.356 | 0.231 | 0.059 | 0.334 | 0.473 |
|  | 40% (5.10B) | 0.572 | 0.287 | 0.526 | 0.185 | 0.367 | 0.214 | 0.260 | 0.440 | 0.347 | 0.231 | 0.021 | 0.314 | 0.444 |
| Ours | 20% (6.55B) | **0.774** | **0.714** | **0.709** | **0.757** | **0.773** | **0.479** | **0.410** | **0.818** | **0.521** | **0.645** | **0.677** | **0.661** | **0.935** |
|  | 30% (5.74B) | **0.749** | **0.651** | **0.658** | **0.553** | **0.687** | **0.412** | **0.372** | **0.776** | **0.483** | **0.501** | **0.629** | **0.588** | **0.832** |
|  | 40% (4.91B) | **0.719** | **0.587** | **0.626** | **0.476** | **0.655** | **0.378** | **0.358** | **0.749** | **0.427** | **0.398** | **0.538** | **0.545** | **0.771** |

Table 2: Performance of the different compression methods on Qwen3-4B-Base. The pretrained backbone and its compressed variants are evaluated on the same set of benchmarks and compression rates as in Table 1. For each compression rate, the best result is shown in **boldface** and the second-best in underlined text.

| Method | CR | PIQA | HS | WG | CSQA | ARC-e | ARC-c | OBQA | boolq | SIQA | mmlu | ld | Avg | RP |
|---|---|---|---|---|---|---|---|---|---|---|---|---|---|---|
| Baseline | 0% (4.02B) | 0.779 | 0.736 | 0.703 | 0.827 | 0.760 | 0.516 | 0.412 | 0.830 | 0.502 | 0.713 | 0.690 | 0.679 | 1.000 |
| Slice GPT | 20% (3.53B) | 0.688 | 0.554 | 0.628 | 0.197 | 0.546 | 0.346 | 0.338 | 0.723 | 0.411 | 0.236 | 0.528 | 0.472 | 0.696 |
|  | 30% (3.06B) | 0.633 | 0.462 | 0.599 | 0.193 | 0.431 | 0.260 | 0.308 | 0.680 | 0.386 | 0.231 | 0.441 | 0.420 | 0.619 |
|  | 40% (2.65B) | 0.584 | 0.384 | 0.553 | 0.197 | 0.348 | 0.251 | 0.276 | 0.602 | 0.371 | 0.230 | 0.359 | 0.378 | 0.556 |
| LaCo | 20% (3.22B) | 0.715 | 0.578 | 0.631 | 0.586 | 0.634 | 0.387 | 0.358 | 0.738 | 0.434 | 0.584 | 0.502 | 0.559 | 0.823 |
|  | 30% (2.81B) | 0.644 | 0.470 | 0.589 | 0.306 | 0.517 | 0.317 | 0.282 | 0.651 | 0.404 | 0.335 | 0.359 | 0.443 | 0.653 |
|  | 40% (2.41B) | 0.630 | 0.416 | 0.562 | 0.195 | 0.453 | 0.273 | 0.284 | 0.606 | 0.388 | 0.234 | 0.341 | 0.398 | 0.587 |
| LLM-Streamline | 20% (3.22B) | **0.739** | 0.559 | 0.556 | 0.196 | 0.619 | 0.369 | 0.378 | 0.558 | 0.417 | 0.235 | 0.448 | 0.461 | 0.679 |
|  | 30% (2.81B) | 0.678 | 0.443 | 0.530 | 0.195 | 0.498 | 0.272 | 0.336 | 0.586 | 0.395 | 0.229 | 0.330 | 0.408 | 0.601 |
|  | 40% (2.41B) | 0.581 | 0.351 | 0.556 | 0.196 | 0.352 | 0.274 | 0.290 | 0.426 | 0.378 | 0.230 | 0.006 | 0.331 | 0.488 |
| Short GPT | 20% (3.22B) | 0.694 | 0.557 | 0.589 | 0.561 | 0.645 | 0.411 | 0.344 | 0.684 | 0.417 | 0.487 | 0.529 | 0.538 | 0.792 |
|  | 30% (2.81B) | 0.654 | 0.386 | 0.551 | 0.185 | 0.492 | 0.308 | 0.312 | 0.588 | 0.372 | 0.245 | 0.253 | 0.395 | 0.582 |
|  | 40% (2.41B) | 0.548 | 0.274 | 0.519 | 0.222 | 0.319 | 0.226 | 0.238 | 0.538 | 0.350 | 0.244 | 0.029 | 0.319 | 0.469 |
| Ours | 20% (3.22B) | 0.736 | **0.662** | **0.669** | **0.779** | **0.704** | **0.436** | **0.382** | **0.784** | **0.501** | **0.657** | **0.651** | **0.633** | **0.932** |
|  | 30% (2.82B) | **0.712** | **0.588** | **0.618** | **0.628** | **0.665** | **0.380** | **0.362** | **0.749** | **0.464** | **0.524** | **0.595** | **0.571** | **0.842** |
|  | 40% (2.41B) | **0.702** | **0.513** | **0.587** | **0.420** | **0.552** | **0.310** | **0.342** | **0.685** | **0.421** | **0.395** | **0.542** | **0.497** | **0.732** |

sequence length of 1,024 tokens, to ensure consistency and robustness in recovered performance across all benchmarks. Comprehensive implementation and experimental details are provided in Appendix B.

## 4.2 RESULTS

We evaluate the proposed compression method on two base LLMs, Qwen3-4B-Base and Qwen3-8B-Base, under compression rates of approximately 20%, 30%, and 40% relative to their original parameter counts. All models were assessed in a zero-shot setting using the LLM evaluation library (Gao et al., 2024a). Additional experiments, including evaluations on other LLM variants and in five-shot settings, are reported in Appendix C.

Tables 2 and 1 summarize the results on compressing Qwen3-4B-Base and Qwen3-8B-Base, respectively. Across all compression rates, our method consistently outperforms competing approaches on most benchmarks, while preserving performance close to that of the uncompressed models. The advantage is most evident on knowledge-intensive tasks such as CSQA, MMLU, and ARC, which rely heavily on retrieving and applying pretrained knowledge. On benchmarks emphasizing common-sense reasoning and general language understanding (e.g., HellaSwag, WinoGrande), the performance gap between methods is smaller, yet our approach still achieves the best overall balance across tasks.

When comparing Qwen3-4B-Base and Qwen3-8B-Base, we observe that the larger base model retains higher absolute accuracy across all compression methods and rates, reflecting its greater capacity. However, the relative performance preservation (RP) of our method remains consistently strong for both model scales, demonstrating its robustness. Notably, the 8B model shows slightly smaller performance degradation under compression, suggesting that larger models may provide more re-

dundancy that can be better exploited during parameter reduction. This trend highlights that while scaling up improves baseline performance, an effective compression strategy is crucial. Overall, our method achieves stable gains across both model sizes, indicating strong generalizability of the approach.

**Discussion.** These findings suggest that updating adapter weights while preserving core model parameters is critical for effective LLM compression. Retaining the pretrained weight structure allows the compressed models to maintain essential knowledge and reasoning capabilities needed for complex tasks. In contrast, methods that aggressively modify core parameters tend to incur larger performance degradation, particularly on knowledge-demanding benchmarks.

## 4.3 ABLATIONS

### 4.3.1 TRAINING STRATEGY

To understand how the size of the adapter groups influences effectiveness and efficiency, we performed an ablation study in which the group size $N$ was varied while keeping all other hyper parameters, compression plan, learning rate schedule, and total training epochs identical to the default configuration described in Appendix B. The experiments, summarized in Table 3, were conducted on Qwen-3-4B-Base compressed to a 20% reduction rate.

When $N = 36$ every adapter is inserted and trained at once, which minimizes the number of training phases but perturbs the entire backbone simultaneously. This large covariate shift leads to unstable gradients and a noticeable drop in downstream performance, as reflected by an average score of 0.6182. At the opposite extreme, $N = 1$ updates one adapter at a time, moving sequentially through the 36 layers. Because only a single component is altered during each step, the original signal is largely preserved, resulting in the highest average performance. However, the training iteration grows roughly linearly with the number of groups, making this setting impractical for larger models.

Table 3: Ablation of the group size $N$ used in the group wise sequential training scheme. The table reports the average downstream score.

| Group size | Avg |
|---|---|
| N=36 (all) | 0.6182 |
| N=1 | 0.6346 |
| N=4 (ours) | 0.6329 |

Our default configuration adopts $N = 4$, grouping four consecutive layers together. This approach retains most of the stability advantages of the single-adapter regime while dramatically reducing the total number of training phases. The resulting average score (0.6329) is only marginally below the optimal $N = 1$ setting, yet the computational cost is comparable to the "all-at-once" baseline. Consequently, we select $N = 4$ as the standard group size for all subsequent experiments.

### 4.3.2 IMPACT OF RECOVERY-TRAINING SET SIZE

Table 4: Effect of recovery-training set size on the performance of our 20% compressed Qwen-3-4B-Base. Results are reported for four different sample budgets (300k, 600k, 1M, and 2M) on a range of downstream benchmarks.

| Method | CR | Samples | PIQA | HS | WG | CSQA | ARC-e | ARC-c | OBQA | boolq | SIQA | mmlu | ld | Avg | RP |
|---|---|---|---|---|---|---|---|---|---|---|---|---|---|---|---|
| Baseline | - | - | 0.7786 | 0.7364 | 0.7032 | 0.8272 | 0.7597 | 0.5162 | 0.4120 | 0.8299 | 0.5015 | 0.7131 | 0.6898 | 0.6789 | 1.0000 |
| Ours | 20% | 300K | 0.7163 | 0.6433 | 0.6630 | 0.7802 | 0.7046 | 0.4181 | 0.3720 | 0.7976 | **0.4928** | 0.6476 | 0.6418 | 0.6252 | 0.9209 |
| | | 600K | 0.7363 | 0.6622 | 0.6690 | 0.7790 | 0.7044 | 0.4358 | 0.3820 | 0.7837 | 0.5013 | 0.6573 | 0.6514 | 0.6329 | 0.9322 |
| | | 1M | 0.7350 | 0.6757 | 0.6788 | **0.8354** | 0.7022 | 0.4488 | 0.3720 | 0.7985 | **0.7084** | 0.6693 | 0.6469 | 0.9528 |
| | | 2M | **0.7679** | **0.7230** | **0.6890** | 0.8215 | **0.7513** | **0.5060** | **0.4020** | **0.8315** | 0.4908 | 0.7076 | **0.6804** | **0.6701** | **0.9870** |

In this section, we evaluate how the size of the recovery-training set influences the effectiveness of our compression pipeline. Table 4 reports results for four different sample budgets (300K, 600K, 1M, and 2M) under a fixed compression rate of 20%. As the number of training instances grows, downstream performance improves consistently across virtually all benchmarks: the average score rises from 0.6252 (300K samples) to 0.6701 (2M samples), and the relative performance (RP) climbs from 0.9209 to 0.9870, narrowing the gap with the uncompressed baseline (Avg=0.6789). For most tasks the improvement is gradual, but a few—namely HS, BoolQ, OBQA, and ARC—show a different pattern. With only 300K–1M samples their scores increase only marginally, reflecting the limited signal provided by a small recovery set. Once the sample count reaches over 1M, the gains accelerate sharply; at 2M samples these tasks almost match the baseline performance (e.g., HS jumps from 0.6757 to 0.7230, BoolQ from 0.7985 to 0.8315, OBQA from 0.3720 to 0.4020, ARC-e from 0.7022 to 0.7513).

Moreover, even the 300K-sample configuration of PnF matches or surpasses strong structured compression baselines such as LaCo, ShortGPT, SliceGPT, and LLM-Streamline that are trained with 600K recovery samples, demonstrating strong data efficiency of PnF.

Overall, the results show that even a modest recovery set captures more than $90\%$ of the attainable relative performance (RP). When the recovery data are scaled to a few million examples, the compressed model nearly matches the uncompressed baseline, incurring less than a $2\%$ performance drop while preserving the $20\%$ compression ratio.

## 5 LIMITATION AND FUTURE WORKS

A possible limitation of our approach is that the first stage of the pipeline is deliberately empirical: the compression plan and grouping schedule currently rely on manually specified per-layer reduction rates and layer groups. While this design grants practitioners flexibility to tailor compression plans to specific deployment constraints, it also places a burden on users to possess a priori knowledge about the relative importance of different layers, which may hinder reproducibility and scalability. In practice, an uninformed choice of layer-wise rates or groups can lead to sub-optimal performance or unnecessary training overhead. On the other hand, this flexibility makes the stage a useful diagnostic tool: by systematically varying the layers or groups that are compressed, users can probe which parts of an LLM are most critical for specific linguistic or reasoning abilities.

Future work will focus on automated, data-driven planning schemes, such as sensitivity-based or reinforcement-learning-based strategies inspired by recent structured pruning and compression methods Wei et al. (2024); Gao et al. (2024b) that learn layer-wise ranks and grouping patterns instead of fixing them heuristically, aiming to reduce manual tuning while preserving the analytical benefits of the current empirical design.

## 6 CONCLUSION

In this work, we introduce a novel framework Plug-and-Fold (PnF), a compression framework that preserves both weights and structure of the pretrained LLM. In our workflow, lightweight PnF adapters are first plugged into a pretrained LLM's weight matrices. After going through adaption phase, adapters are folded back into the base model via simple matrix multiplication. The resulting model is structurally identical to the original backbone yet enjoys substantial reductions in parameters with unimpaired performance. Extensive experiments on four backbones and three compression rates show PnF consistently outperforms strong baselines, highlighting the benefit of retaining pretrained weights. Ablation studies on training strategies confirm the effectiveness of our workflow, while experiments on recovery-training set size demonstrate that with sufficient data PnF can nearly match the original model's performance. In summary, Plug-and-Fold provides an efficient, scalable, architecture-preserving compression pipeline that maintains the expressive power of large pretrained LLMs, enabling deployment on resource-constrained hardware without performance loss.

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

# A   PnF COMPRESSION PIPELINE (PSEUDO-CODE)

Algorithm 1 provides a concise pseudo-code summary of the plug-and-fold (PnF) pipeline, including (i) compression planning of per-layer rank allocation and selection of layers to compress, (ii) group-wise adapter training on a frozen backbone, and (iii) the folding step that replaces each compressed layer's weight with folded weight $W_l^{\text{Comp}} = W_l P_l$.

---

**Algorithm 1** Plug-and-Fold (PnF) Compression Pipeline

---

**Require:** Pretrained decoder-only LLM weights $\{W_l\}_{l=1}^L$, global compression rate $\rho$, teacher model $f_{\text{teacher}}$, distillation dataset $\mathcal{D}$
**Ensure:** Compressed weights $\{W_l^{\text{Comp}}\}_{l=1}^L$
1: **[Compression Planning]**
2: Compute per-layer target ranks/widths $\{r_l\}_{l=1}^L$ to match the global compression rate $\rho$.
3: Define the set of layers to compress $\mathcal{C} \leftarrow \{l \in \{1, \ldots, L\} : r_l < \text{full\_dim}(W_l)\}$.
4: Partition $\mathcal{C}$ into groups $G_1, \ldots, G_K$ (single-stage PnF uses $K{=}1$ and $G_1{=}\mathcal{C}$).
5: **[Group-Wise Sequential Training for adapter]**
6: Initialize PnF adapters $\{P_l\}_{l \in \mathcal{C}}$ (e.g., near-identity).
7: **for** $k = 1$ to $K$ **do**
8:     **for** each layer $l \in G_k$ **do**
9:         Attach adapter $P_l$ to $W_l$ (replace $xW_l$ by $xW_lP_l$ with rank $r_l$).
10:         Freeze $W_l$ and mark only $P_l$ as trainable.
11:     **end for**
12:     **for** training step $t = 1$ to $T_k$ **do**
13:         Sample a minibatch $x \sim \mathcal{D}$.
14:         Compute teacher outputs $p_{\text{teacher}}(\cdot \mid x) = f_{\text{teacher}}(x)$.
15:         Compute student outputs $p_{\text{student}}(\cdot \mid x; \{W_l, P_l\})$.
16:         Update $\{P_l\}_{l \in G_k}$ by minimizing a distillation loss (e.g., $\text{KL}(p_{\text{teacher}} \| p_{\text{student}})$).
17:     **end for**
18:     **[Folding Step for adapter (for saving memory in training process)]**
19:     **for** each layer $l \in G_k$ **do**
20:         $W_l^{\text{Comp}} \leftarrow W_l P_l$.
21:         Remove $P_l$ and keep $W_l^{\text{Comp}}$ for inference.
22:     **end for**
23: **end for**

---

# B   EXPERIMENT SETTINGS

## B.1   HYPER-PARAMETER CONFIGURATION

In all experiments we follow the two-stage pipeline described in Section 3.2. Below we detail the hyperparameter settings that were used to instantiate the compression plan, to construct the training groups, and to train the adapters. The values are the same for every model and compression rate unless explicitly noted. Also, the PnF are initialized as identity matrix, where only the diagonal elements are set to 1 otherwise 0.

## B.2   COMPRESSION PLAN (PER-LAYER REDUCTION RATES)

For each target compression rate $c \in \{20\%, 30\%, 40\%\}$, we empirically driven target hidden-dimension targets for the multi-head self-attention (MHSA) and feed-forward network (FFN) sub-layers. The resulting dimensionalities are listed in Table 5. The notation indicates the target hidden size for each group in the order in which the groups are visited (from the output side toward the input side).

**Interpretation of Table 5** Taking OPT 2.7B as an example, for a $20\%$ reduction the first two groups (closest to the output) compress both the MHSA projection matrices to $r_{\text{head}} = 72$ and the

Table 5: Target hidden dimensions for MHSA and FFN at each compression rate. Each entry corresponds to a successive group of layers (see Figure 2).

| Backbone | CR | MHSA | FFN |
|---|---|---|---|
| OPT 2.7B | 20% | $(72, 72, -, -)$ | $(3584, 3584, 4096, 4864)$ |
| | 30% | $(64, 72, 72, -, -, -)$ | $(3328, 3328, 3840, 4608, 5632, 6144, 8192)$ |
| | 40% | $(64, 64, 72, 72, -, -, -, -)$ | $(2560, 2560, 2816, 2816, 2816, 3840, 5888, 8096)$ |
| OPT 6.7B | 20% | $(-, -, -, -)$ | $(5120, 5632, 7168, 7168)$ |
| | 30% | $(64, 80, 96, 112, -, -)$ | $(4608, 5376, 6144, 8192, 10240, 13312)$ |
| | 40% | $(64, 64, 64, 64, 96, -)$ | $(5632, 5376, 5120, 5120, 7168, 7168)$ |
| Qwen3 4B | 20% | $(-, -, -, -, -)$ | $(2560, 2816, 3328, 4608, 9216)$ |
| | 30% | $(-, -, -, -, -, -)$ | $(2560, 2560, 2560, 3072, 3584, 4864)$ |
| | 40% | $(-, -, -, -, -, -, -, -)$ | $(2560, 2560, 2560, 2816, 2816, 3072, 3328, 5632)$ |
| Qwen3 8B | 20% | $(-, -, -, -, -)$ | $(4096, 4352, 4864, 6144, 8704)$ |
| | 30% | $(-, -, -, -, -, -, -, -)$ | $(4096, 4352, 4608, 4864, 4864, 5632, 7680)$ |
| | 40% | $(-, -, -, -, -, -, -, -, -, -)$ | $(4096, 4352, 4608, 4608, 4352, 4608, 4608, 4608, 7936)$ |

FFN intermediate dimensions to $r_{\text{inter}} = 3584$. Subsequent groups use the next values in the list, while " $-$ " denotes it retains the original dimension. At 30% and 40% the plan contains more groups, thereby spreading the reduction more gradually across the stack.

### B.3 TRAINING SCHEDULE

The overall workflow of training is as follows. For each selected group $G_k$ we:

1. Insert PnF adapters corresponding to index belonging to $G_k$
2. Train for $E$ epochs while keeping all previously trained groups frozen
3. Proceed to $G_{k+1}$ until $G_{n_g}$

Through out the entire experiments, the number of epochs is fixed to $E := 1$, giving a total of $n_g$ iteration.

### B.4 FOLDING STEP

After the final group has been trained, each adapter pair is merged into its corresponding projection matrix $W$ by the closed-form multiplication. No additional fine-tuning is performed after folding, which guarantees that the resulting model has exactly the same architecture and runtime characteristics as the original uncompressed model.

### B.5 BASELINE RECOVERY FINE-TUNING SETTINGS

For the recovery-fine-tuning (RFT) stage we adopt LoRA, since LoRA fine-tuning is widely used in recent work. To ensure a fair comparison, we fix the low-rank dimension to $r = 16$ for every LoRA experiment. Unless a particular method explicitly restricts its scope, LoRA is applied to all transformer layers—both the multi-head self-attention (MHSA) and feed-forward network (FFN) sub-layers.

## C ADDITIONAL RESULTS

### C.1 COMPARISON WITH BASELINE METHODS

In this section we compare our proposed approach with several baselines across a broader set of conditions. We evaluate four backbone models—Qwen-3-4B-Base, Qwen-3-8B-Base, OPT-2.7B, and OPT-6.7B, LLaMA-3.2-3B, LLaMA-3.1-8B-and we assess performance in both zero-shot and five-shot settings. Across all experiments, our method consistently yields the highest average score (Avg), closely matching the performance of the uncompressed baseline for each backbone.

The same trend observed in the zero-shot experiments holds in the five-shot setting. Our compression method consistently outperforms the baselines across all compression rates, and the performance gap widens on knowledge-intensive benchmarks. Thus, the superior performance of our approach is preserved when a few exemplars are provided.

Table 6: Performance of the different compression methods on LLaMA-3.2-3B on zero-shots setting.

| Method | CR | Sample | PIQA | HS | WG | CSQA | ARC-e | ARC-c | OBQA | boolq | SIQA | mmlu | ld | Avg | RP |
|---|---|---|---|---|---|---|---|---|---|---|---|---|---|---|---|
| Baseline | 0% (3.21B) | - | 0.7748 | 0.7370 | 0.6906 | 0.6404 | 0.7168 | 0.4582 | 0.4320 | 0.7278 | 0.4708 | 0.5396 | 0.7000 | 0.6262 | – |
| Slice GPT | 20% (2.90B) | 600k | 0.5664 | 0.3318 | 0.5217 | 0.1998 | 0.3396 | 0.2543 | 0.3020 | 0.5966 | 0.3552 | 0.2420 | 0.0714 | 0.3437 | – |
| | 30% (2.56B) | | 0.5484 | 0.3178 | 0.4996 | 0.1966 | 0.3211 | 0.2415 | 0.2580 | 0.5841 | 0.3449 | 0.2562 | 0.0732 | 0.3310 | – |
| | 40% (2.22B) | | 0.5424 | 0.2923 | 0.4980 | 0.1957 | 0.3026 | 0.2278 | 0.2540 | 0.4798 | 0.3444 | 0.2617 | 0.0472 | 0.3133 | – |
| LaCo | 20% (2.61B) | 600k | 0.7002 | 0.6330 | 0.6890 | 0.6183 | 0.6044 | 0.3771 | 0.3520 | 0.6697 | 0.4427 | 0.5177 | 0.6330 | 0.5670 | – |
| | 30% (2.21B) | | 0.6736 | 0.5134 | 0.5864 | 0.3227 | 0.5248 | 0.3131 | 0.3200 | 0.6242 | 0.4033 | 0.3148 | 0.4801 | 0.4615 | – |
| | 40% (1.90B) | | 0.6028 | 0.4156 | 0.5667 | 0.2228 | 0.3952 | 0.2491 | 0.2840 | 0.6217 | 0.3915 | 0.2652 | 0.3972 | 0.4011 | – |
| LLM-Streamline | 20% (2.61B) | 600k | 0.7138 | 0.6171 | 0.6661 | 0.6372 | 0.6103 | 0.3840 | 0.3740 | 0.7150 | 0.4401 | 0.5450 | 0.5131 | 0.5651 | – |
| | 30% (2.21B) | | 0.6763 | 0.5317 | 0.6504 | 0.4390 | 0.5459 | 0.3345 | 0.3160 | 0.6450 | 0.4150 | 0.4172 | 0.4147 | 0.4896 | – |
| | 40% (1.90B) | | 0.6556 | 0.3884 | 0.5162 | 0.1949 | 0.4743 | 0.2517 | 0.3000 | 0.6076 | 0.3608 | 0.2295 | 0.1906 | 0.3791 | – |
| Short GPT | 20% (2.61B) | 600k | 0.6948 | 0.6095 | 0.6827 | 0.6126 | 0.5947 | 0.3840 | 0.3520 | 0.6419 | 0.4473 | 0.5207 | 0.6043 | 0.5586 | – |
| | 30% (2.21B) | | 0.6425 | 0.4954 | 0.6369 | 0.5315 | 0.4769 | 0.3123 | 0.3080 | 0.6355 | 0.4115 | 0.4806 | 0.3831 | 0.4831 | – |
| | 40% (1.90B) | | 0.6110 | 0.3929 | 0.5809 | 0.1949 | 0.3948 | 0.2713 | 0.2860 | 0.6226 | 0.3675 | 0.2299 | 0.2243 | 0.3797 | – |
| Ours | 20% (2.55B) | 600k | 0.7548 | 0.6541 | 0.6877 | 0.6073 | 0.6824 | 0.4029 | 0.3820 | 0.6868 | 0.4430 | 0.5087 | 0.6493 | 0.5872 | – |
| | 30% (2.22B) | | 0.7331 | 0.5916 | 0.6256 | 0.4998 | 0.6283 | 0.3625 | 0.3500 | 0.6798 | 0.4235 | 0.4386 | 0.5981 | 0.5392 | – |
| | 40% (1.92B) | | 0.6977 | 0.5169 | 0.5830 | 0.3500 | 0.5436 | 0.3041 | 0.3480 | 0.6391 | 0.4087 | 0.3571 | 0.5294 | 0.4798 | – |

Table 7: Performance of the different compression methods on LLaMA-3.2-3B on five-shots setting.

| Method | PIQA | HS | WG | CSQA | ARC-e | ARC-c | OBQA | boolq | SIQA | mmlu | lambada_openai |
|---|---|---|---|---|---|---|---|---|---|---|---|
| Baseline 0% (3.21B) | 0.8025 | 0.7546 | 0.7238 | 0.6658 | 0.7816 | 0.4838 | 0.4489 | 0.7336 | 0.5066 | 0.5616 | 0.6652 |
| Slice GPT 20% (2.90B) | 0.5805 | 0.3358 | 0.5359 | 0.1892 | 0.3573 | 0.2415 | 0.2640 | 0.6031 | 0.3634 | 0.2539 | 0.0505 |
| Slice GPT 30% (2.56B) | 0.5528 | 0.3216 | 0.5257 | 0.1925 | 0.3430 | 0.2389 | 0.2760 | 0.5355 | 0.3414 | 0.2516 | 0.0611 |
| Slice GPT 40% (2.22B) | 0.5365 | 0.2921 | 0.5154 | 0.1867 | 0.3148 | 0.2355 | 0.2620 | 0.4321 | 0.3347 | 0.2519 | 0.0380 |
| LaCo 20% (2.61B) | 0.7095 | 0.6463 | 0.6890 | 0.6486 | 0.6616 | 0.4019 | 0.3500 | 0.6951 | 0.4846 | 0.5219 | 0.5913 |
| LaCo 30% (2.21B) | 0.6823 | 0.5281 | 0.5991 | 0.2678 | 0.5694 | 0.3311 | 0.3240 | 0.6217 | 0.4417 | 0.3109 | 0.3798 |
| LaCo 40% (1.90B) | 0.6104 | 0.4173 | 0.5841 | 0.2154 | 0.4205 | 0.2551 | 0.2840 | 0.6135 | 0.3869 | 0.2559 | 0.3767 |
| LLM-Streamline 20% (2.61B) | 0.7236 | 0.6373 | 0.6827 | 0.6536 | 0.6561 | 0.3891 | 0.3600 | 0.7428 | 0.4826 | 0.5548 | 0.4564 |
| LLM-Streamline 30% (2.21B) | 0.6828 | 0.5475 | 0.6709 | 0.4120 | 0.5829 | 0.3473 | 0.3220 | 0.6755 | 0.4478 | 0.4190 | 0.3546 |
| LLM-Streamline 40% (1.90B) | 0.6545 | 0.3870 | 0.5178 | 0.2097 | 0.4718 | 0.2423 | 0.2800 | 0.5410 | 0.3639 | 0.2465 | 0.1679 |
| Short GPT 20% (2.61B) | 0.6964 | 0.6372 | 0.6875 | 0.6396 | 0.6414 | 0.3831 | 0.3560 | 0.6673 | 0.4821 | 0.5394 | 0.5845 |
| Short GPT 30% (2.21B) | 0.6507 | 0.5114 | 0.6433 | 0.5536 | 0.5130 | 0.3097 | 0.3140 | 0.6315 | 0.4386 | 0.4657 | 0.3910 |
| Short GPT 40% (1.90B) | 0.6094 | 0.3970 | 0.5714 | 0.1957 | 0.4196 | 0.2696 | 0.2760 | 0.6064 | 0.3838 | 0.2553 | 0.2199 |
| Ours 20% (2.55B) | 0.7742 | 0.6719 | 0.6896 | 0.6274 | 0.7238 | 0.4281 | 0.3920 | 0.7338 | 0.4975 | 0.5231 | 0.6056 |
| Ours 30% (2.22B) | 0.7409 | 0.6297 | 0.6461 | 0.5144 | 0.6486 | 0.3889 | 0.3700 | 0.6906 | 0.4806 | 0.4686 | 0.5553 |
| Ours 40% (1.92B) | 0.7175 | 0.5390 | 0.6083 | 0.3727 | 0.5623 | 0.3223 | 0.3600 | 0.6599 | 0.4606 | 0.3785 | 0.4904 |

Table 8: Performance of the different compression methods on LLaMA-3.1-8B on zero-shots setting.

| Method | CR | Sample | PIQA | HS | WG | CSQA | ARC-e | ARC-c | OBQA | boolq | SIQA | mmlu | ld | Avg | RP |
|---|---|---|---|---|---|---|---|---|---|---|---|---|---|---|---|
| Baseline | 0% (8.03B) | - | 0.8123 | 0.7884 | 0.7356 | 0.7150 | 0.8123 | 0.5367 | 0.4460 | 0.8196 | 0.4713 | 0.6345 | 0.7533 | 0.6841 | 1.0000 |
| Slice GPT | 20% (6.41B) | 600k | 0.5582 | 0.3818 | 0.5414 | 0.2015 | 0.3068 | 0.2449 | 0.2840 | 0.5535 | 0.3511 | 0.2466 | 0.0726 | 0.3402 | 0.4973 |
| | 30% (5.61B) | | 0.5854 | 0.3592 | 0.5335 | 0.1966 | 0.3603 | 0.2568 | 0.2760 | 0.4590 | 0.3403 | 0.2376 | 0.0819 | 0.3351 | 0.4898 |
| | 40% (4.83B) | | 0.5609 | 0.3299 | 0.5067 | 0.1957 | 0.3439 | 0.2321 | 0.2560 | 0.4367 | 0.3454 | 0.2461 | 0.0770 | 0.3210 | 0.4692 |
| LaCo | 20% (6.50B) | 600k | 0.7693 | 0.7056 | 0.6875 | 0.5209 | 0.7155 | 0.4317 | 0.3800 | 0.7691 | 0.4565 | 0.4671 | 0.6534 | 0.5961 | 0.8714 |
| | 30% (5.63B) | | 0.7280 | 0.6209 | 0.6630 | 0.3604 | 0.6233 | 0.3558 | 0.3500 | 0.6667 | 0.4350 | 0.3478 | 0.5694 | 0.5200 | 0.7601 |
| | 40% (4.76B) | | 0.6670 | 0.5141 | 0.6243 | 0.4210 | 0.5139 | 0.3055 | 0.2900 | 0.6312 | 0.4181 | 0.4141 | 0.4809 | 0.4800 | 0.7017 |
| LLM-Streamline | 20% (6.50B) | 600k | 0.7514 | 0.7007 | 0.7238 | 0.6912 | 0.7214 | 0.4633 | 0.3940 | 0.7609 | 0.4585 | 0.6164 | 0.3872 | 0.6062 | 0.8861 |
| | 30% (5.63B) | | 0.6986 | 0.6035 | 0.6906 | 0.7035 | 0.6170 | 0.3763 | 0.3620 | 0.7593 | 0.4360 | 0.6271 | 0.3949 | 0.5699 | 0.8331 |
| | 40% (4.76B) | | 0.6785 | 0.4778 | 0.5872 | 0.1941 | 0.5059 | 0.2833 | 0.3260 | 0.6190 | 0.4007 | 0.2301 | 0.2750 | 0.4161 | 0.6082 |
| Short GPT | 20% (6.50B) | 600k | 0.7465 | 0.6924 | 0.7159 | 0.6986 | 0.7024 | 0.4437 | 0.3740 | 0.7214 | 0.4611 | 0.5919 | 0.7075 | 0.6232 | 0.9110 |
| | 30% (5.63B) | | 0.6855 | 0.5914 | 0.6993 | 0.5872 | 0.5875 | 0.3626 | 0.3060 | 0.7113 | 0.4222 | 0.4208 | 0.5410 | 0.5377 | 0.7860 |
| | 40% (4.76B) | | 0.6213 | 0.4531 | 0.5983 | 0.1974 | 0.4310 | 0.2807 | 0.2780 | 0.6226 | 0.3889 | 0.2302 | 0.3427 | 0.4040 | 0.5906 |
| Ours | 20% (6.42B) | 600k | 0.7750 | 0.7369 | 0.7130 | 0.6326 | 0.7494 | 0.4654 | 0.4040 | 0.7942 | 0.4606 | 0.6170 | 0.7226 | 0.6428 | 0.9396 |
| | 30% (5.62B) | | 0.7584 | 0.7048 | 0.6932 | 0.5892 | 0.7149 | 0.4389 | 0.3920 | 0.7525 | 0.4503 | 0.5713 | 0.6942 | 0.6145 | 0.8983 |
| | 40% (4.48B) | | 0.7439 | 0.6240 | 0.6438 | 0.4562 | 0.6135 | 0.3494 | 0.3620 | 0.7094 | 0.4368 | 0.4528 | 0.6219 | 0.5467 | 0.7992 |

Table 9: Performance of the different compression methods on LLaMA-3.1-8B on five-shots setting.

| Method | CR | Sample | PIQA | HS | WG | CSQA | ARC-e | ARC-c | OBQA | boolq | SIQA | mmlu | ld | Avg | RP |
|---|---|---|---|---|---|---|---|---|---|---|---|---|---|---|---|
| Baseline | 0% (8.03B) | - | 0.8243 | 0.8092 | 0.7719 | 0.7412 | 0.8502 | 0.5768 | 0.4640 | 0.8275 | 0.5251 | 0.6503 | 0.6848 | 0.7023 | – |
| Slice GPT | 20% (6.41B) | | 0.5756 | 0.4066 | 0.5541 | 0.1916 | 0.3359 | 0.2491 | 0.3020 | 0.6355 | 0.3675 | 0.2339 | 0.0660 | 0.3562 | – |
| | 30% (5.61B) | 600k | 0.6023 | 0.3787 | 0.5320 | 0.1990 | 0.4398 | 0.2773 | 0.2520 | 0.5410 | 0.2385 | 0.0681 | | 0.3535 | – |
| | 40% (4.83B) | | 0.5740 | 0.3350 | 0.5217 | 0.2113 | 0.3826 | 0.2432 | 0.2620 | 0.4330 | 0.3582 | 0.2465 | 0.0530 | 0.3291 | – |
| LaCo | 20% (6.50B) | | 0.7688 | 0.7243 | 0.6953 | 0.5356 | 0.7437 | 0.4539 | 0.3860 | 0.7639 | 0.4985 | 0.4490 | 0.6140 | 0.6030 | – |
| | 30% (5.63B) | 600k | 0.7274 | 0.6413 | 0.6630 | 0.4357 | 0.6810 | 0.3891 | 0.3560 | 0.7131 | 0.4724 | 0.4323 | 0.4830 | 0.5449 | – |
| | 40% (4.76B) | | 0.6654 | 0.5273 | 0.6243 | 0.4595 | 0.5581 | 0.3038 | 0.3120 | 0.6636 | 0.4437 | 0.4385 | 0.4407 | 0.4943 | – |
| LLM-Streamline | 20% (6.50B) | | 0.7563 | 0.7269 | 0.7648 | 0.7314 | 0.7626 | 0.4804 | 0.4080 | 0.8080 | 0.5164 | 0.6339 | 0.3553 | 0.6313 | – |
| | 30% (5.63B) | 600k | 0.6828 | 0.5475 | 0.6709 | 0.4120 | 0.5829 | 0.3473 | 0.3220 | 0.6755 | 0.4478 | 0.4190 | 0.3546 | 0.4966 | – |
| | 40% (4.76B) | | 0.6545 | 0.3870 | 0.5178 | 0.2097 | 0.4718 | 0.2423 | 0.2800 | 0.5410 | 0.3639 | 0.2465 | 0.1679 | 0.3711 | – |
| Short GPT | 20% (6.50B) | | 0.7508 | 0.7202 | 0.7388 | 0.7281 | 0.7462 | 0.4676 | 0.3720 | 0.6596 | 0.5118 | 0.6318 | 0.6231 | 0.6318 | – |
| | 30% (5.63B) | 600k | 0.6942 | 0.6085 | 0.7230 | 0.6298 | 0.6237 | 0.3737 | 0.3240 | 0.7095 | 0.4821 | 0.4828 | 0.5300 | 0.5619 | – |
| | 40% (4.76B) | | 0.6328 | 0.4622 | 0.6212 | 0.2031 | 0.4735 | 0.2935 | 0.2860 | 0.6214 | 0.4222 | 0.2793 | 0.3113 | 0.4188 | – |
| Ours | 20% (6.42B) | | 0.8094 | 0.7548 | 0.7482 | 0.6714 | 0.7990 | 0.5132 | 0.4300 | 0.8046 | 0.5131 | 0.6262 | 0.6641 | 0.6667 | – |
| | 30% (5.62B) | 600k | 0.7844 | 0.7224 | 0.7082 | 0.6063 | 0.7662 | 0.4680 | 0.4140 | 0.7656 | 0.4969 | 0.5842 | 0.6462 | 0.6329 | – |
| | 40% (4.48B) | | 0.7527 | 0.6369 | 0.6546 | 0.4776 | 0.6818 | 0.3754 | 0.3760 | 0.7332 | 0.4772 | 0.4761 | 0.5814 | 0.5657 | – |

Table 10: Performance of the different compression methods on Qwen3-4B-Base on five-shots setting.

| Method | CR | Sample | PIQA | HS | WG | CSQA | ARC-e | ARC-c | OBQA | boolq | SIQA | mmlu | ld | Avg | RP |
|---|---|---|---|---|---|---|---|---|---|---|---|---|---|---|---|
| Baseline | 0% (4.02B) | - | 0.7889 | 0.7532 | 0.7206 | 0.8198 | 0.8674 | 0.6425 | 0.4500 | 0.8654 | 0.5502 | 0.7319 | 0.6501 | 0.7127 | 1.0000 |
| Slice GPT | 20% (3.53B) | | 0.6980 | 0.5612 | 0.6425 | 0.3030 | 0.6902 | 0.4130 | 0.3480 | 0.7746 | 0.4641 | 0.3250 | 0.4487 | 0.5153 | 0.7230 |
| | 30% (3.06B) | 600k | 0.6409 | 0.4661 | 0.6085 | 0.2293 | 0.5370 | 0.2952 | 0.3120 | 0.6911 | 0.4181 | 0.2651 | 0.3656 | 0.4390 | 0.6160 |
| | 40% (2.65B) | | 0.5832 | 0.3857 | 0.5596 | 0.1925 | 0.4158 | 0.2440 | 0.2780 | 0.511 | 0.3909 | 0.2672 | 0.2928 | 0.3746 | 0.5256 |
| LaCo | 20% (3.22B) | | 0.7236 | 0.5840 | 0.6425 | 0.7273 | 0.7016 | 0.4249 | 0.3680 | 0.7679 | 0.4698 | 0.6192 | 0.4496 | 0.5889 | 0.8264 |
| | 30% (2.81B) | 600k | 0.6398 | 0.475 | 0.5841 | 0.3194 | 0.5556 | 0.3362 | 0.2820 | 0.7028 | 0.4252 | 0.2863 | 0.3043 | 0.4464 | 0.6264 |
| | 40% (2.41B) | | 0.6300 | 0.4136 | 0.5509 | 0.2080 | 0.4996 | 0.2944 | 0.2880 | 0.6242 | 0.4083 | 0.2810 | 0.2550 | 0.4048 | 0.5680 |
| LLM-Streamline | 20% (3.22B) | | 0.7448 | 0.5572 | 0.5241 | 0.2015 | 0.7428 | 0.4292 | 0.3880 | 0.5474 | 0.4544 | 0.2895 | 0.3974 | 0.4797 | 0.6730 |
| | 30% (2.81B) | 600k | 0.6724 | 0.4333 | 0.5059 | 0.1891 | 0.5883 | 0.3054 | 0.3180 | 0.6012 | 0.4027 | 0.2538 | 0.3049 | 0.4159 | 0.5836 |
| | 40% (2.41B) | | 0.5865 | 0.3468 | 0.5643 | 0.1957 | 0.3742 | 0.2611 | 0.2800 | 0.3841 | 0.3602 | 0.2295 | 0.0060 | 0.3262 | 0.4577 |
| Short GPT | 20% (3.22B) | | 0.7008 | 0.5520 | 0.6014 | 0.5766 | 0.7189 | 0.4573 | 0.3280 | 0.6914 | 0.4631 | 0.5167 | 0.4644 | 0.5519 | 0.7743 |
| | 30% (2.81B) | 600k | 0.6088 | 0.3142 | 0.5138 | 0.1974 | 0.4196 | 0.2747 | 0.2480 | 0.3847 | 0.3561 | 0.2446 | 0.0134 | 0.3250 | 0.4561 |
| | 40% (2.41B) | | 0.5294 | 0.2564 | 0.4972 | 0.2080 | 0.2950 | 0.2568 | 0.2460 | 0.3869 | 0.3439 | 0.2370 | 0.0000 | 0.2961 | 0.4154 |
| Ours | 20% (3.22B) | | 0.7559 | 0.6714 | 0.6772 | 0.8003 | 0.7739 | 0.4955 | 0.4100 | 0.8355 | 0.5417 | 0.6771 | 0.6055 | 0.6585 | 0.9240 |
| | 30% (2.82B) | 600k | 0.7233 | 0.5847 | 0.6343 | 0.6798 | 0.7070 | 0.4008 | 0.4000 | 0.7602 | 0.4955 | 0.5412 | 0.5411 | 0.5880 | 0.8250 |
| | 40% (2.41B) | | 0.6912 | 0.5134 | 0.5783 | 0.5030 | 0.6186 | 0.3487 | 0.3540 | 0.7283 | 0.4517 | 0.3956 | 0.4757 | 0.5144 | 0.7218 |

Table 11: Performance of the different compression methods on Opt 6.7B in zero-shot setting.

| Method | CR | Sample | PIQA | HS | WG | CSQA | ARC-e | ARC-c | OBQA | boolq | SIQA | mmlu | ld | Avg | RP |
|---|---|---|---|---|---|---|---|---|---|---|---|---|---|---|---|
| Baseline | 0% (6.66B) | - | 0.7644 | 0.6719 | 0.6543 | 0.2031 | 0.6002 | 0.3473 | 0.3760 | 0.6612 | 0.4278 | 0.2505 | 0.6769 | 0.5121 | 1.0000 |
| Slice GPT | 20% (5.49B) | | 0.7165 | 0.5657 | 0.6204 | 0.1916 | 0.5055 | 0.2961 | 0.3560 | 0.6235 | 0.4206 | 0.2500 | 0.5632 | 0.4645 | 0.9070 |
| | 30% (4.77B) | 600k | 0.7013 | 0.5220 | 0.6093 | 0.1957 | 0.4735 | 0.2875 | 0.3320 | 0.6064 | 0.3976 | 0.2421 | 0.4890 | 0.4415 | 0.8621 |
| | 40% (4.07B) | | 0.6589 | 0.4709 | 0.5604 | 0.1982 | 0.4495 | 0.2671 | 0.3280 | 0.5835 | 0.3899 | 0.2290 | 0.4017 | 0.4125 | 0.8054 |
| LaCo | 20% (5.25B) | | 0.6866 | 0.5310 | 0.6014 | 0.2064 | 0.4899 | 0.2995 | 0.3280 | 0.6214 | 0.4165 | 0.2503 | 0.5088 | 0.4491 | 0.8769 |
| | 30% (4.64B) | 600k | 0.6213 | 0.3890 | 0.5446 | 0.1974 | 0.3965 | 0.2560 | 0.2980 | 0.6214 | 0.3735 | 0.2463 | 0.1764 | 0.3746 | 0.7315 |
| | 40% (4.04B) | | 0.5930 | 0.3391 | 0.5170 | 0.1957 | 0.3481 | 0.2363 | 0.2740 | 0.6211 | 0.3613 | 0.2371 | 0.0638 | 0.3442 | 0.6722 |
| LLM-Streamline | 20% (5.25B) | | 0.7361 | 0.6037 | 0.6172 | 0.1761 | 0.5745 | 0.3191 | 0.3320 | 0.6324 | 0.4165 | 0.2470 | 0.5492 | 0.4731 | 0.9238 |
| | 30% (4.64B) | 600k | 0.6953 | 0.4204 | 0.5588 | 0.1974 | 0.5198 | 0.2850 | 0.3260 | 0.6330 | 0.3904 | 0.2381 | 0.2791 | 0.4130 | 0.8065 |
| | 40% (4.04B) | | 0.6284 | 0.3430 | 0.5288 | 0.1966 | 0.4491 | 0.2304 | 0.2960 | 0.6217 | 0.3464 | 0.2311 | 0.1186 | 0.3627 | 0.7083 |
| Short GPT | 20% (5.25B) | | 0.5044 | 0.2597 | 0.5051 | 0.1957 | 0.2668 | 0.2594 | 0.2720 | 0.3783 | 0.3515 | 0.2295 | 0.0000 | 0.2929 | 0.5720 |
| | 30% (4.64B) | 600k | 0.5065 | 0.2578 | 0.4917 | 0.1957 | 0.2597 | 0.2568 | 0.2860 | 0.3783 | 0.3418 | 0.2295 | 0.0000 | 0.2913 | 0.5687 |
| | 40% (4.04B) | | 0.5065 | 0.2579 | 0.4878 | 0.1957 | 0.2601 | 0.2491 | 0.2980 | 0.3783 | 0.3454 | 0.2295 | 0.0000 | 0.2917 | 0.5695 |
| Ours | 20% (5.32B) | | 0.7403 | 0.6126 | 0.6461 | 0.2146 | 0.5886 | 0.3278 | 0.3600 | 0.6666 | 0.4207 | 0.2567 | 0.6135 | 0.4952 | 0.9671 |
| | 30% (4.66B) | 600k | 0.7126 | 0.5321 | 0.6127 | 0.1998 | 0.5495 | 0.3069 | 0.3340 | 0.6496 | 0.4140 | 0.2512 | 0.5269 | 0.4627 | 0.9035 |
| | 40% (3.99B) | | 0.6417 | 0.4926 | 0.5920 | 0.1966 | 0.4877 | 0.2874 | 0.3260 | 0.6382 | 0.3949 | 0.2464 | 0.4728 | 0.4342 | 0.8479 |

Table 12: Performance of the different compression methods on Opt 2.7B in zero-shot setting.

| Method | CR | Sample | PIQA | HS | WG | CSQA | ARC-e | ARC-c | OBQA | boolq | SIQA | mmlu | ld | Avg | RP |
|---|---|---|---|---|---|---|---|---|---|---|---|---|---|---|---|
| Baseline | 0% (2.65B) | - | 0.7481 | 0.6063 | 0.6101 | 0.1990 | 0.5438 | 0.3131 | 0.3520 | 0.6027 | 0.4212 | 0.2567 | 0.6361 | 0.4808 | 1.0000 |
| Slice GPT | 20% (2.23B) | | 0.6654 | 0.4682 | 0.5904 | 0.2031 | 0.4322 | 0.2637 | 0.3300 | 0.5257 | 0.3838 | 0.2415 | 0.4108 | 0.4104 | 0.8537 |
| | 30% (1.94B) | 600k | 0.6300 | 0.4228 | 0.5635 | 0.1966 | 0.4175 | 0.2585 | 0.3060 | 0.5168 | 0.3705 | 0.2316 | 0.3551 | 0.3881 | 0.8072 |
| | 40% (1.66B) | | 0.5865 | 0.3674 | 0.5343 | 0.1957 | 0.3742 | 0.2509 | 0.2820 | 0.3982 | 0.3602 | 0.2301 | 0.2880 | 0.3516 | 0.7313 |
| LaCo | 20% (2.10B) | | 0.6697 | 0.4629 | 0.5612 | 0.1957 | 0.4356 | 0.2782 | 0.3080 | 0.6223 | 0.3899 | 0.2436 | 0.4768 | 0.4222 | 0.8781 |
| | 30% (1.86B) | 600k | 0.6197 | 0.3677 | 0.5627 | 0.2113 | 0.3699 | 0.2415 | 0.2880 | 0.5832 | 0.3853 | 0.2330 | 0.1469 | 0.3645 | 0.7581 |
| | 40% (1.63B) | | 0.5762 | 0.3006 | 0.5193 | 0.1957 | 0.3308 | 0.2261 | 0.2920 | 0.5920 | 0.3561 | 0.2312 | 0.0279 | 0.3316 | 0.6897 |
| LLM-Streamline | 20% (2.10B) | | 0.7100 | 0.5471 | 0.6038 | 0.1974 | 0.5097 | 0.2867 | 0.3240 | 0.6058 | 0.4053 | 0.2537 | 0.5692 | 0.4557 | 0.9478 |
| | 30% (1.86B) | 600k | 0.6763 | 0.4016 | 0.5438 | 0.1966 | 0.4609 | 0.2585 | 0.3160 | 0.6012 | 0.3756 | 0.2344 | 0.2876 | 0.3957 | 0.8230 |
| | 40% (1.63B) | | 0.6023 | 0.3122 | 0.5114 | 0.1949 | 0.3788 | 0.2150 | 0.2760 | 0.6119 | 0.3454 | 0.2298 | 0.0778 | 0.3414 | 0.7101 |
| Short GPT | 20% (2.10B) | | 0.6692 | 0.4476 | 0.5745 | 0.1941 | 0.4457 | 0.2696 | 0.3080 | 0.5929 | 0.3904 | 0.2315 | 0.3155 | 0.4035 | 0.8393 |
| | 30% (1.86B) | 600k | 0.5354 | 0.2715 | 0.5083 | 0.1982 | 0.3081 | 0.2381 | 0.2600 | 0.3789 | 0.3459 | 0.2301 | 0.0029 | 0.2979 | 0.6197 |
| | 40% (1.63B) | | 0.5152 | 0.2677 | 0.5067 | 0.1974 | 0.2908 | 0.2500 | 0.2600 | 0.3810 | 0.3423 | 0.2315 | 0.0035 | 0.2951 | 0.6138 |
| Ours | 20% (2.11B) | | 0.7235 | 0.5012 | 0.6088 | 0.2023 | 0.5139 | 0.2922 | 0.3460 | 0.6287 | 0.4243 | 0.2500 | 0.5666 | 0.4598 | 0.9563 |
| | 30% (1.85B) | 600k | 0.6908 | 0.4615 | 0.5741 | 0.1981 | 0.4724 | 0.2782 | 0.3180 | 0.6157 | 0.4132 | 0.2462 | 0.5407 | 0.4347 | 0.9042 |
| | 40% (1.58B) | | 0.6642 | 0.4205 | 0.5449 | 0.1957 | 0.4486 | 0.2759 | 0.2940 | 0.5861 | 0.4020 | 0.2388 | 0.4584 | 0.4117 | 0.8564 |

Table 13: Performance of the different compression methods on Opt 6.7B in five-shot setting.

| Method | CR | Sample | PIQA | HS | WG | CSQA | ARC-e | ARC-c | OBQA | boolq | SIQA | mmlu | ld | Avg | RP |
|---|---|---|---|---|---|---|---|---|---|---|---|---|---|---|---|
| Baseline | 0% (6.66B) | - | 0.7704 | 0.6797 | 0.6598 | 0.1867 | 0.6982 | 0.3703 | 0.3920 | 0.7012 | 0.4785 | 0.2634 | 0.6451 | 0.5314 | 1.0000 |
| Slice GPT | 20% (5.49B) | | 0.7187 | 0.5652 | 0.6211 | 0.1981 | 0.5984 | 0.3293 | 0.3600 | 0.5492 | 0.4206 | 0.2622 | 0.4189 | 0.4583 | 0.8625 |
| | 30% (4.77B) | 600k | 0.6921 | 0.5221 | 0.6314 | 0.1826 | 0.5699 | 0.3063 | 0.3280 | 0.5318 | 0.4124 | 0.2553 | 0.3623 | 0.4358 | 0.8202 |
| | 40% (4.07B) | | 0.6561 | 0.4669 | 0.5912 | 0.1859 | 0.5173 | 0.2790 | 0.3220 | 0.5028 | 0.3935 | 0.2666 | 0.2925 | 0.4067 | 0.7654 |
| LaCo | 20% (5.25B) | | 0.6915 | 0.5318 | 0.6069 | 0.2146 | 0.5244 | 0.3038 | 0.3280 | 0.6217 | 0.4355 | 0.2595 | 0.4935 | 0.4556 | 0.8573 |
| | 30% (4.64B) | 600k | 0.6170 | 0.3914 | 0.5375 | 0.1998 | 0.4411 | 0.2730 | 0.2840 | 0.6220 | 0.3817 | 0.2549 | 0.1300 | 0.3757 | 0.7069 |
| | 40% (4.04B) | | 0.5919 | 0.3399 | 0.5312 | 0.1949 | 0.3733 | 0.2406 | 0.2660 | 0.6211 | 0.3541 | 0.2542 | 0.0324 | 0.3454 | 0.6500 |
| LLM-Streamline | 20% (5.25B) | | 0.7426 | 0.6207 | 0.5943 | 0.2006 | 0.6485 | 0.3455 | 0.3700 | 0.6519 | 0.4600 | 0.2522 | 0.5356 | 0.4929 | 0.9275 |
| | 30% (4.64B) | 600k | 0.6219 | 0.3529 | 0.5099 | 0.1810 | 0.4428 | 0.2338 | 0.2640 | 0.5927 | 0.3572 | 0.2496 | 0.0714 | 0.3525 | 0.6633 |
| | 40% (4.04B) | | 0.5811 | 0.2982 | 0.4964 | 0.1998 | 0.3577 | 0.2167 | 0.2560 | 0.5838 | 0.3326 | 0.2433 | 0.0213 | 0.3261 | 0.6136 |
| Short GPT | 20% (5.25B) | | 0.5060 | 0.2606 | 0.5233 | 0.1957 | 0.2622 | 0.2594 | 0.2680 | 0.3783 | 0.3490 | 0.2295 | 0.0000 | 0.2938 | 0.5529 |
| | 30% (4.64B) | 600k | 0.4984 | 0.2562 | 0.4957 | 0.1957 | 0.2563 | 0.2474 | 0.2800 | 0.3783 | 0.3423 | 0.2295 | 0.0000 | 0.2891 | 0.5440 |
| | 40% (4.04B) | | 0.5054 | 0.2552 | 0.4972 | 0.1957 | 0.2546 | 0.2534 | 0.2680 | 0.3783 | 0.3464 | 0.2295 | 0.0000 | 0.2907 | 0.5470 |
| Ours | 20% (5.32B) | | 0.7647 | 0.6255 | 0.6319 | 0.2080 | 0.6477 | 0.3423 | 0.3720 | 0.6729 | 0.4683 | 0.2610 | 0.6032 | 0.5089 | 0.9576 |
| | 30% (4.66B) | 600k | 0.7323 | 0.5273 | 0.6221 | 0.1909 | 0.5905 | 0.3167 | 0.3520 | 0.6461 | 0.4468 | 0.2547 | 0.5081 | 0.4716 | 0.8874 |
| | 40% (3.99B) | | 0.6896 | 0.4673 | 0.6038 | 0.1959 | 0.5343 | 0.2819 | 0.3320 | 0.6086 | 0.4292 | 0.2501 | 0.3951 | 0.4353 | 0.8191 |

Table 14: Performance of the different compression methods on Opt 2.7B in five-shot setting.

| Method | CR | Sample | PIQA | HS | WG | CSQA | ARC-e | ARC-c | OBQA | boolq | SIQA | mmlu | ld | Avg | RP |
|---|---|---|---|---|---|---|---|---|---|---|---|---|---|---|---|
| Baseline | 0% (2.65B) | - | 0.7481 | 0.6068 | 0.6204 | 0.1884 | 0.6469 | 0.3311 | 0.3580 | 0.6272 | 0.4550 | 0.2579 | 0.6010 | 0.4946 | 1.0000 |
| Slice GPT | 20% (2.23B) | | 0.6757 | 0.4632 | 0.5770 | 0.1933 | 0.5080 | 0.2918 | 0.3100 | 0.4205 | 0.4099 | 0.2457 | 0.3037 | 0.3999 | 0.8085 |
| | 30% (1.94B) | 600k | 0.6322 | 0.4179 | 0.5746 | 0.2015 | 0.4609 | 0.2551 | 0.3000 | 0.4477 | 0.3991 | 0.2538 | 0.2663 | 0.3826 | 0.7736 |
| | 40% (1.66B) | | 0.5936 | 0.3612 | 0.5383 | 0.2080 | 0.3880 | 0.2449 | 0.2800 | 0.4349 | 0.3756 | 0.2480 | 0.1974 | 0.3518 | 0.7113 |
| LaCo | 20% (2.10B) | | 0.6746 | 0.4600 | 0.5825 | 0.1925 | 0.4886 | 0.2824 | 0.2900 | 0.6217 | 0.4252 | 0.2628 | 0.4221 | 0.4275 | 0.8643 |
| | 30% (1.86B) | 600k | 0.6186 | 0.3690 | 0.5588 | 0.1900 | 0.3986 | 0.2491 | 0.2600 | 0.6211 | 0.3705 | 0.2465 | 0.1025 | 0.3622 | 0.7324 |
| | 40% (1.63B) | | 0.5745 | 0.2973 | 0.5130 | 0.2023 | 0.3350 | 0.2287 | 0.2600 | 0.6208 | 0.3561 | 0.2366 | 0.0155 | 0.3309 | 0.6690 |
| LLM-Streamline | 20% (2.10B) | | 0.7198 | 0.5554 | 0.6006 | 0.1990 | 0.5871 | 0.3012 | 0.3260 | 0.6000 | 0.4385 | 0.2512 | 0.4925 | 0.4610 | 0.9321 |
| | 30% (1.86B) | 600k | 0.6436 | 0.4228 | 0.5138 | 0.1818 | 0.4524 | 0.2627 | 0.2720 | 0.5422 | 0.3689 | 0.2570 | 0.2327 | 0.3773 | 0.7628 |
| | 40% (1.63B) | | 0.5539 | 0.2823 | 0.5075 | 0.1990 | 0.3338 | 0.2099 | 0.2500 | 0.5673 | 0.3336 | 0.2505 | 0.0165 | 0.3186 | 0.6441 |
| Short GPT | 20% (2.10B) | | 0.6442 | 0.4013 | 0.5604 | 0.1916 | 0.4566 | 0.2637 | 0.2980 | 0.5621 | 0.3935 | 0.2505 | 0.1970 | 0.3835 | 0.7754 |
| | 30% (1.86B) | 600k | 0.5152 | 0.2553 | 0.5257 | 0.1966 | 0.2727 | 0.2457 | 0.2800 | 0.3783 | 0.3413 | 0.2295 | 0.0000 | 0.2946 | 0.5956 |
| | 40% (1.63B) | | 0.5011 | 0.2572 | 0.5193 | 0.2007 | 0.2685 | 0.2820 | 0.2820 | 0.3783 | 0.3413 | 0.2342 | 0.0000 | 0.2953 | 0.5970 |
| Ours | 20% (2.11B) | | 0.7107 | 0.5642 | 0.6099 | 0.1901 | 0.5835 | 0.3101 | 0.3320 | 0.6300 | 0.4302 | 0.2534 | 0.5110 | 0.4659 | 0.9420 |
| | 30% (1.85B) | 600k | 0.6794 | 0.4540 | 0.5741 | 0.2015 | 0.5243 | 0.2894 | 0.3300 | 0.5701 | 0.4291 | 0.2588 | 0.4518 | 0.4330 | 0.8754 |
| | 40% (1.58B) | | 0.6518 | 0.4096 | 0.5551 | 0.1966 | 0.4827 | 0.2777 | 0.3080 | 0.5498 | 0.4230 | 0.2503 | 0.3678 | 0.4066 | 0.8220 |

Table 15: Performance of the different compression methods on Qwen3-8B-Base with five shots setting. The pretrained backbone and its compressed variants are evaluated on the same set of benchmarks and compression rates as in Table 1.

| Method | CR | PIQA | HS | WG | CSQA | ARC-e | ARC-c | OBQA | boolq | SIQA | mmlu | ld | Avg | RP |
|---|---|---|---|---|---|---|---|---|---|---|---|---|---|---|
| Baseline | 0% (8.19B) | 0.815 | 0.795 | 0.770 | 0.856 | 0.880 | 0.681 | 0.490 | 0.882 | 0.572 | 0.770 | 0.671 | 0.744 | 1.000 |
| | 20% (6.52B) | 0.714 | 0.632 | 0.686 | 0.329 | 0.747 | 0.462 | 0.396 | 0.781 | 0.496 | 0.356 | 0.527 | 0.557 | 0.749 |
| Slice GPT | 30% (5.71B) | 0.676 | 0.553 | 0.642 | 0.275 | 0.621 | 0.361 | 0.370 | 0.696 | 0.443 | 0.275 | 0.456 | 0.488 | 0.656 |
| | 40% (4.91B) | 0.627 | 0.451 | 0.594 | 0.201 | 0.494 | 0.279 | 0.318 | 0.614 | 0.415 | 0.255 | 0.363 | 0.419 | 0.564 |
| | 20% (6.65B) | 0.736 | 0.651 | 0.671 | 0.709 | 0.748 | 0.493 | 0.406 | 0.534 | 0.503 | 0.604 | 0.546 | 0.600 | 0.807 |
| LaCo | 30% (5.88B) | 0.694 | 0.535 | 0.600 | 0.506 | 0.629 | 0.358 | 0.318 | 0.673 | 0.456 | 0.408 | 0.471 | 0.514 | 0.690 |
| | 40% (5.10B) | 0.617 | 0.403 | 0.572 | 0.215 | 0.487 | 0.297 | 0.276 | 0.623 | 0.402 | 0.251 | 0.256 | 0.400 | 0.538 |
| | 20% (6.65B) | 0.774 | 0.613 | 0.561 | 0.238 | 0.769 | 0.446 | 0.402 | 0.548 | 0.477 | 0.268 | 0.462 | 0.505 | 0.680 |
| LLM-Streamline | 30% (5.88B) | 0.724 | 0.500 | 0.553 | 0.194 | 0.673 | 0.338 | 0.346 | 0.450 | 0.418 | 0.243 | 0.310 | 0.432 | 0.580 |
| | 40% (5.10B) | 0.608 | 0.364 | 0.568 | 0.196 | 0.392 | 0.266 | 0.310 | 0.451 | 0.382 | 0.230 | 0.010 | 0.343 | 0.462 |
| | 20% (6.65B) | 0.574 | 0.301 | 0.494 | 0.197 | 0.353 | 0.260 | 0.252 | 0.592 | 0.346 | 0.247 | 0.003 | 0.329 | 0.443 |
| Short GPT | 30% (5.88B) | 0.561 | 0.278 | 0.494 | 0.195 | 0.327 | 0.227 | 0.252 | 0.493 | 0.347 | 0.256 | 0.002 | 0.312 | 0.420 |
| | 40% (5.10B) | 0.540 | 0.258 | 0.512 | 0.198 | 0.307 | 0.230 | 0.256 | 0.417 | 0.348 | 0.229 | 0.000 | 0.300 | 0.403 |
| | 20% (6.55B) | 0.788 | 0.718 | 0.730 | 0.796 | 0.819 | 0.540 | 0.442 | 0.853 | 0.546 | 0.660 | 0.653 | 0.686 | 0.922 |
| Ours | 30% (5.74B) | 0.753 | 0.654 | 0.679 | 0.658 | 0.737 | 0.442 | 0.392 | 0.786 | 0.511 | 0.501 | 0.574 | 0.608 | 0.817 |
| | 40% (4.91B) | 0.724 | 0.587 | 0.637 | 0.526 | 0.680 | 0.402 | 0.352 | 0.775 | 0.463 | 0.404 | 0.487 | 0.549 | 0.738 |

# D ON KNOWLEDGE PRESERVATION VIA ADAPTER FOLDING

This section provides additional evidence for the claim that adapter folding better preserves pre-trained knowledge than conventional low-rank compression. We present (i) an empirical comparison between adapt-before-folding and train-after-folding pipelines under matched rank and training budget, and (ii) an intuitive small-scale example that clarifies why learned adapters can retain more of the original transformation than truncation-first strategies.

## D.1 EMPIRICAL COMPARISON OF RECONSTRUCTION FIDELITY

We compare our default **PnF pipeline** (adapt-before-folding) with the **baseline** (train-after-folding) under the same compression ratio (20%) and the same training budget (600K samples). In the baseline, the model is first reduced to the target rank $r$ (i.e., $r_{\text{head}}$, $r_{\text{inter}}$) using a low-rank projector, and the resulting compressed weights are directly fine-tuned without any adapters. In PnF, the pretrained backbone is frozen, only lightweight adapters are trained with KL-distillation, and the adapters are folded into a dense matrix after training. Table 16 reports zero-shot performance for Qwen-3-4B-Base compressed to 20% across a diverse set of benchmarks. At a fixed rank and training budget, PnF consistently outperforms the train-after-folding baseline on most tasks as well as on the averaged metric, indicating that adapter-based parameterization preserves the pretrained model's behavior more effectively than directly training on truncated weights.

Table 16: Comparison between baseline (train-after-folding) and PnF (adapt-before-folding) at 20% compression on Qwen-3-4B-Base. All models are trained for 600K samples.

| Method | Comp.(%) | PIQA | HS | WG | CSQA | ARC-e | ARC-c | OBQA | BoolQ | SIQA | MMLU | LD | Avg |
|---|---|---|---|---|---|---|---|---|---|---|---|---|---|
| Baseline | 20% | 0.704 | 0.636 | 0.642 | 0.740 | 0.675 | 0.416 | 0.334 | 0.772 | 0.473 | 0.607 | 0.621 | 0.602 |
| PnF | 20% | 0.736 | 0.662 | 0.669 | 0.779 | 0.704 | 0.435 | 0.382 | 0.783 | 0.501 | 0.657 | 0.651 | 0.632 |

## D.2 TOY EXAMPLE: TRUNCATION-FIRST VS. ADAPT-BEFORE-FOLDING

To build intuition, we contrast the backbone transformation, a truncation-first strategy, and our adapt-before-folding scheme on a simple $2 \times 3$ example.

Let

$$x = \begin{bmatrix} a \\ b \end{bmatrix}, \qquad W = \begin{bmatrix} c & d & e \\ f & g & h \end{bmatrix} \in \mathbb{R}^{2 \times 3}.$$

**Backbone.** The original pretrained transformation is

$$f_W(x) = x^\top W = \begin{bmatrix} a & b \end{bmatrix} \begin{bmatrix} c & d & e \\ f & g & h \end{bmatrix} = \begin{bmatrix} ac + bf & ad + bg & ae + bh \end{bmatrix}.$$

**Truncation-first.** A truncation-first strategy applies a fixed selector $S \in \mathbb{R}^{3 \times 2}$ that drops the third column, e.g.

$$S = \begin{bmatrix} 1 & 0 \\ 0 & 1 \\ 0 & 0 \end{bmatrix}, \qquad W^{\text{Trunc}} = WS = \begin{bmatrix} c & d \\ f & g \end{bmatrix}.$$

The compressed transformation becomes

$$f_{W^{\text{Trunc}}}(x) = x^\top W^{\text{Trunc}} = \begin{bmatrix} a & b \end{bmatrix} \begin{bmatrix} c & d \\ f & g \end{bmatrix} = \begin{bmatrix} ac + bf & ad + bg \end{bmatrix}.$$

Here the contribution of the third column $(e, h)$ is discarded by construction; the compressed model can only exploit the first two columns of $W$.

**Adapt-before-folding (PnF).** In PnF, we instead introduce a learnable adapter $P \in \mathbb{R}^{3 \times 2}$ and keep $W$ frozen. For illustration, write

$$P = \begin{bmatrix} i & j \\ k & l \\ m & n \end{bmatrix}, \qquad W^{\text{Comp}} = WP.$$

Then

$$f_{W^{\text{Comp}}}(x) = x^\top W^{\text{Comp}} = x^\top WP = \begin{bmatrix} a & b \end{bmatrix} \begin{bmatrix} c & d & e \\ f & g & h \end{bmatrix} \begin{bmatrix} i & j \\ k & l \\ m & n \end{bmatrix}$$

$$= \left[ (ac + bf)i + (ad + bg)k + (ae + bh)m \quad (ac + bf)j + (ad + bg)l + (ae + bh)n \right].$$

Although $W^{\text{Comp}}$ also has only two output dimensions, every entry of the original $W$ (including $e$ and $h$) participates in the product $WP$ and can still influence $f_{W^{\text{Comp}}}(x)$ through the learned coefficients $(i, j, k, l, m, n)$. The adapter $P$ is optimized so that $f_{W^{\text{Comp}}}(x) \approx f_W(x)$ on the training distribution, effectively redistributing the contribution of all columns of $W$ into the lower-dimensional representation. This toy example highlights the qualitative difference between truncation-first and adapt-before-folding: truncation irrevocably removes part of the pretrained weights, whereas PnF retains the full pretrained matrix and learns how to compress it via a data-driven adapter.

## E  TRAINING COST AND PRACTICAL EFFICIENCY OF PNF

In this section, we provide the experimental results with a cost-performance analysis of PnF. Table 17 reports training latency and average downstream zero-shot performance for Qwen-3-4B-Base compressed to 20% on an $8 \times$H100 configuration under two training regimes: a single-stage non-sequential variant, where all adapters are trained at once, and the group-wise sequential schedule used in our main experiments. At a compression ratio (20%) and training budget (600K samples), non-sequential (all-at-once) PnF uses a single-stage training loop similar in structure to standard fine-tuning, yet it already recovers strong performance surpasses the baselines (LaCo: 0.559 vs. PnF: 0.602). By contrast, group-wise sequential training increases recovery phase by roughly a factor of three, but yields a clear additional gain in average performance ($0.602 \rightarrow 0.633$). Importantly, the proposed group-wise sequential training schedule is optional for our method to work rather than a requirement of PnF: the non-sequential "all-at-once" PnF training already provides a competitive cost–performance trade-off, and practitioners can choose the configuration that best fits their resources, using the single-stage variant when wall-clock time is limited and the multi-stage variant when the highest possible performance is desired.

Table 17: Training latency and average performance for the two PnF training regimes on Qwen-3-4B-Base (20% compression, 600K samples, $8 \times$H100).

| Regime | # Groups | Training latency (h) | GPU hours | Avg. performance |
|---|---|---|---|---|
| Non-sequential training | 1 | 8.21 | 65.7 | 0.602 |
| Sequential training | 4 | 23.32 | 186.6 | 0.633 |

## F  ADDITIONAL COMPARISONS WITH SVD-LLM AND BITDISTILLER

We provide additional baseline comparisons to SVD-LLM(Wang et al., 2024) and BitDistiller (Du et al., 2024) at a common 20% compression ratio on LLaMA-2-7B, in order to offer a more comprehensive and fair evaluation against structured low-rank and quantization/distillation baselines.

### F.1 COMPARISON WITH SVD-LLM

Following the experimental setup of the original SVD-LLM paper, we use LLaMA-2-7B and evaluate PnF at a 20% compression ratio on the shared tasks: PIQA, HellaSwag, WinoGrande, ARC-e, OpenBookQA, GSM8K, MathQA, and TruthfulQA. The results for PnF are obtained under our standard PnF training pipeline, and the numbers for SVD-LLM are taken from (Wang et al., 2024) under the same compression ratio and backbone. At this 20% setting, PnF attains a higher average performance than SVD-LLM.

Table 18: Performance comparison between PnF and SVD-LLM on LLaMA-2-7B at 20% compression.

| Method | PIQA | HellaSwag | WinoGrande | ARC-e | OpenBookQA | GSM8K | MathQA | TruthfulQA | Avg. |
|---|---|---|---|---|---|---|---|---|---|
| SVD-LLM | 0.69 | 0.52 | 0.68 | 0.59 | 0.33 | 0.08 | 0.26 | 0.28 | 0.43 |
| PnF | 0.76 | 0.56 | 0.69 | 0.73 | 0.33 | 0.09 | 0.26 | 0.38 | 0.48 |

### F.2 COMPARISON WITH BITDISTILLER

BitDistiller (Du et al., 2024) is a low-bit quantization framework that integrates quantization-aware training with self-distillation, whereas PnF focuses on structured compression. Because BitDistiller applies only quantization, exact model-size matching with our method, which uses structured compression, is difficult. Nonetheless, we adopt the same backbone (LLaMA-2-7B) and compare the performance of BitDistiller with 3-bit quantization to PnF with 20% compression rate with additional 4 bit quantization. Even under this conservative setting (quantization-only vs. structured compression + quantization), PnF+quant. achieves higher or comparable downstream performance than the 3-bit BitDistiller model.

Table 19: Benchmark comparison between BitDistiller and PnF with 4-bit quantization on LLaMA-2-7B.

| Method | PIQA | HellaSwag | WinoGrande | ARC-c | MMLU | Avg. |
|---|---|---|---|---|---|---|
| BitDistiller | 0.7699 | 0.5538 | 0.6835 | 0.4121 | 0.4465 | 0.5732 |
| PnF + quant. | 0.7673 | 0.5645 | 0.6941 | 0.4184 | 0.4285 | 0.5746 |

## G ADDITIONAL COMPARISONS WITH LLM-PRUNER AND DISP-LLM

We provide additional baseline comparisons with LLM-Pruner (Ma et al., 2023) and DISP-LLM (Gao et al., 2024b) at a 20% compression ratio on LLaMA-2-7B, to offer a more comprehensive evaluation against latest structured compression methods.

### G.1 COMPARISON WITH LLM-PRUNER

For a fair comparison, we align with the experimental setting of LLM-Pruner by adopting LLaMA-2-7B and comparing results at a 20% compression ratio on the shared benchmarks (PIQA, HellaSwag, WinoGrande, ARC-e, ARC-c, OpenBookQA, and BoolQ). The results for LLM-Pruner are sourced from (Ma et al., 2023), and our PnF results are obtained under the same backbone and compression ratio. Under this setting, as summarized in Table 20, PnF attains a higher average performance than LLM-Pruner.

### G.2 COMPARISON WITH DISP-LLM

We additionally compare PnF with DISP-LLM (Gao et al., 2024b), another latest structured compression approach. Using LLaMA-2-7B and the same 20% compression setting, we place our PnF results alongside the DISP-LLM performance reported in (Gao et al., 2024b) on the overlapping benchmarks. PnF again achieves higher average performance.

Table 20: Benchmark comparison between LLM-Pruner and PnF on LLaMA-2-7B at 20% compression.

| Method | PIQA | HellaSwag | WinoGrande | ARC-e | ARC-c | OpenBookQA | BoolQ | Avg. |
|--------|------|-----------|------------|-------|-------|------------|-------|------|
| LLM-Pruner | 0.76 | 0.68 | 0.65 | 0.63 | 0.38 | 0.40 | 0.70 | 0.60 |
| PnF (LLM-Pruner) | 0.78 | 0.70 | 0.69 | 0.73 | 0.41 | 0.42 | 0.75 | 0.62 |

Table 21: Benchmark comparison between DISP-LLM and PnF on LLaMA-2-7B at 20% compression. "–" denotes a missing value.

| Method | PIQA | HellaSwag | WinoGrande | ARC-e | ARC-c | Avg. |
|--------|------|-----------|------------|-------|-------|------|
| DISP-LLM | 0.77 | 0.68 | 0.65 | 0.65 | 0.37 | 0.62 |
| PnF (DISP-LLM) | 0.78 | 0.70 | 0.69 | 0.73 | 0.41 | 0.66 |

## H  INFERENCE LATENCY ANALYSIS

We present supplemental results analyzing the practical impact of compression on generation speed. We measure the average per-token generation latency (in milliseconds) for the original backbone, SliceGPT (Ashkboos et al., 2024), and PnF on Qwen3-4B-Base using a single H100 GPU under different compression ratios (0%, 20%, 30%, and 40%). As shown in Table 22, the average per-token latency decreases as the compression ratio increases for both PnF and SliceGPT.

Table 22: Average per-token generation latency (ms) of SliceGPT and PnF at different compression ratios on Qwen3-4B-Base with $1\times$H100. Here, 0% means that the model is not compressed.

| Method | 0% | 20% | 30% | 40% |
|--------|------|------|------|------|
| SliceGPT | 29.126 ms | 22.299 ms | 22.120 ms | 22.099 ms |
| PnF | 29.126 ms | 22.081 ms | 21.719 ms | 21.559 ms |

## I  COMPATIBILITY WITH LOW-BIT QUANTIZATION

To evaluate the deployment efficacy of PnF, we additionally examine whether the folded matrices, $W^{\text{Comp}} = WP$, produced by PnF introduce distributional shifts that complicate low-bit quantization compared to the original weight. To this end, we apply the same 4-bit post-training quantization pipeline to both the uncompressed Qwen3-4B-Base model and its PnF-compressed variants at 20%, 30%, and 40% compression ratios. The results are summarized in Table 23. For the uncompressed baseline, 4-bit quantization reduces the average score from 0.6789 to 0.6441, a 5.13% relative drop. For PnF, the corresponding drops are $0.6329 \rightarrow 0.6244$ (1.13%), $0.5713 \rightarrow 0.5454$ (4.53%), and $0.4971 \rightarrow 0.4720$ (5.05%), for 20%, 30%, and 40% compression, respectively. In other words, the quantization-induced degradation for PnF model does not exceed that of full-size backbone. These results provide empirical evidence that the folded weights, $W^{\text{Comp}}$, do not introduce harmful outliers that would harm the quantization pipeline, and, therefore, fully compatible with standard 4-bit post-training quantization pipelines.

Table 23: 4-bit post-training quantization results on Qwen3-4B-Base and its PnF-compressed variants. "Drop rate" denotes the relative performance drop compared to the corresponding FP16 model.

| Method | Comp. Ratio | PIQA | HS | WG | CSQA | ARC-e | ARC-c | OBQA | BoolQ | SIQA | MMLU | LAMBADA | Avg | Drop rate |
|---|---|---|---|---|---|---|---|---|---|---|---|---|---|---|
| Baseline | 0% (4.02B) | 0.7786 | 0.7364 | 0.7032 | 0.8272 | 0.7597 | 0.5162 | 0.4120 | 0.8299 | 0.5015 | 0.7131 | 0.6898 | 0.6789 | 0 |
| Baseline (4bit) | 0% (4.02B) | 0.7758 | 0.7200 | 0.6811 | 0.7721 | 0.7214 | 0.4906 | 0.3940 | 0.8086 | 0.4918 | 0.6774 | 0.5525 | 0.6441 | 5.13% (↓) |
| Ours | 20% (3.22B) | 0.7363 | 0.6622 | 0.6690 | 0.7790 | 0.7044 | 0.4358 | 0.3820 | 0.7837 | 0.5013 | 0.6573 | 0.6514 | 0.6329 | 0 |
| Ours | 30% (2.82B) | 0.7118 | 0.5878 | 0.6177 | 0.6275 | 0.6649 | 0.3803 | 0.3620 | 0.7485 | 0.4639 | 0.5243 | 0.5951 | 0.5713 | 0 |
| Ours | 40% (2.41B) | 0.7015 | 0.5134 | 0.5874 | 0.4195 | 0.5517 | 0.3095 | 0.3420 | 0.6846 | 0.4210 | 0.3949 | 0.5423 | 0.4971 | 0 |
| Ours (4bit) | 20% (3.22B) | 0.7341 | 0.6436 | 0.6611 | 0.7424 | 0.6704 | 0.4221 | 0.3660 | 0.7632 | 0.4839 | 0.6209 | 0.5549 | 0.6244 | 1.13% (↓) |
| Ours (4bit) | 30% (2.82B) | 0.7142 | 0.5698 | 0.6091 | 0.5759 | 0.6489 | 0.3760 | 0.3520 | 0.7054 | 0.4481 | 0.5002 | 0.5000 | 0.5454 | 4.53% (↓) |
| Ours (4bit) | 40% (2.41B) | 0.6874 | 0.4985 | 0.5743 | 0.4021 | 0.5349 | 0.2969 | 0.3160 | 0.6547 | 0.4063 | 0.3765 | 0.4439 | 0.4720 | 5.05% (↓) |

## J    STATEMENT OF LARGE-LANGUAGE-MODEL (LLM) USAGE

The authors acknowledge that a large-language-model (LLM) was employed as a general-purpose assistance tool during the preparation of this manuscript. Specifically, the following tasks were supported by the LLM under the direct supervision of the authors:

- Formatting and LaTeX assistance – The LLM supplied LaTeX snippets for tables, equations, and figure captions (e.g., Table 5 and the hyper-parameter description). The authors integrated these snippets into the manuscript and performed all final compilation and formatting checks.
- Language polishing – The LLM was used to improve readability, correct grammar, and adjust stylistic tone across the entire manuscript. The final wording reflects the authors' own decisions after thorough review.

All content generated by the LLM was fully supervised, fact-checked, and substantially revised by the human authors before inclusion in the final version. No portion of the manuscript was submitted to the LLM for autonomous generation without subsequent author verification.

The authors affirm that the intellectual contributions, experimental design, data analysis, and conclusions are entirely their own work, and that the LLM served only as an auxiliary writing and editing aid.

