# OpenReview forum: "Plug-and-Fold: Weight-Preserving Structured Compression for Large Language Models"
_ICLR.cc/2026/Conference — Submitted to ICLR 2026_

### Official Review · Reviewer_HJXN · 2025-10-25

**Soundness:** 2
**Presentation:** 2
**Contribution:** 2
**Rating:** 4
**Confidence:** 3

**Summary:**

The paper presents  a structured compression method for LLMs that mitigates the performance loss of pruning. PnF inserts lightweight, low-rank adapters into frozen linear layers of a pre-trained model. These adapters are trained using a KL-divergence distillation loss and a Group-wise Sequential Training (GWST) scheme, then folded into the base weights by simple matrix multiplication. The result is a dimension-reduced model with the same architecture and zero runtime overhead. Experiments show consistent gains over strong baselines, though the approach relies on manual compression schedules and lacks full efficiency analysis.

**Strengths:**

+ The method provides a clean separation between the knowledge locked in the pre-trained weights (frozen $W$) and the structural compression process (learning $P$). This is an elegant theoretical step forward from disruptive structured pruning techniques.

+ The post-training folding step is a crucial practical advantage. By fully merging the adapters into the base weights, the method avoids introducing any new operations or latency during inference, ensuring the compressed model is truly efficient and deployment-friendly.

+ The performance gains over strong compression baselines (SliceGPT, LaCo, etc.) are notable and consistent across different Qwen and OPT model scales. This suggests the core principle of PnF is highly effective at preserving model fidelity during dimensionality reduction.

**Weaknesses:**

+ PnF’s uses a hand-crafted per-layer reduction schedule, with no automatic or data-driven metric, thereby limiting reproducibility and generalization.

+ Although framed as an efficiency approach, the paper reports only parameter counts. No FLOPs or latency data are provided.

+ Recovering near-baseline performance requires large amount of distillation samples

+ No Comparison to Analytical Low-Rank Methods:

**Questions:**

+ Could you provide quantitative evidence (e.g., activation similarity or representation distance metrics) showing how freezing W
contributes to knowledge retention compared to jointly training W and P?

+ Have you examined whether the folded matrices W^Comp =W⋅P introduce non-Gaussian outliers or distributional shifts that could complicate low-bit quantization (e.g., 4-bit) compared to quantizing the original
W?

---

> ### Author Response · Authors · 2025-11-21
> **Response to Weakness 1**
>
> We thnk the reviewer HJXN for thoroughly reviewing our work. Hope these responses can clarify the reviewer's concerns.
>
>
>
> We acknowledge that the design of the per-layer reduction schedule is important for reproducibility and generalization.
>
> - **Reproducibility**: Our pyramidal per-layer schedule is indeed hand-designed, but it is fully specified and fixed: for each model and compression ratio, we use a deterministic rule (later layers compressed more aggressively), and we reuse the same rule across all runs without tuning. The exact layer-wise ratios are already provided in Appendix. We will make this pointer more prominent so that the schedule is easy to reproduce.
>
> - **Generalization**: We intentionally **apply the same simple schedule** across different decoder-only LLMs (OPT, Qwen, LLaMA-3) and multiple compression ratios (e.g., 20/30/40%), and still outperform the compression baselines (e.g., SliceGPT, LLM-Streamline) under all these settings. This empirical robustness across architectures and compression ratios suggests that our results are not tied to a brittle, heavily tuned schedule, even though more sophisticated data-driven allocation schemes could potentially further improve performance.
>
> - **Limitation and Future work**: We acknowledge that the current PnF implementation relies on a heuristic schedule for assigning per‑layer ranks, which we already note as a limitation in the manuscript. We view this as orthogonal to the core framework, which only requires per-layer ranks and can be combined with automatic allocation strategies. Automated planning based on sensitivity or reinforcement learning is a promising way to improve generality and reduce manual tuning; in fact, we are already exploring such data-driven and RL-style schemes (learning layer-wise ranks instead of fixing them heuristically), and we will add this as a concrete future direction in the limitation/discussion section.

---

> > ### Author Response · Authors · 2025-11-21
> > **Response to Weakness 2**
> >
> > We understand that efficiency must be measured in more than just parameter counts. To this end, we have added Table 1, which reports the average per‑token inference latency for the uncompressed Qwen3-4B backbone and its PnF-compressed counterpart (i.e., 20%, 30\%, and 40\%). All measurements were obtained under a fixed hardware and batch/sequence configuration (1×H100, batch size 1, sequence length 128). The results show that the PnF consistently reduces per-token latency as the compression ratio increases. In the revised manuscript, we will incorporate this table into the main text and discuss it explicitly as evidence of efficiency, also describing how per-token latency varies across different compression settings.
> >
> > **Table 1.** Latency per-token generation of SliceGPT vs. our method at different compression ratios.
> >
> > | Method   | 0% (No comp) | 20%      | 30%      | 40%      |
> > |----------|--------------|----------|----------|----------|
> > | SliceGPT | 29.126 ms    | 22.299 ms| 22.120 ms| 22.099 ms|
> > | Ours     | 29.126 ms    | 22.081 ms| 21.719 ms| 21.559 ms|

---

> > > ### Author Response · Authors · 2025-11-21
> > > **Response to Weakness 3**
> > >
> > > We also see the recovery budget as an important practical consideration. In our experiments, however, **all methods use the same recovery dataset (e.g., 600K samples)**, and under this shared budget, **PnF consistently outperforms all compression baselines** while retaining a high fraction of the baseline performance.
> > >
> > > Moreover, **PnF remains competitive even at tighter budgets: with 300K samples, it already exceeds the performance of other methods trained on 600K samples**, demonstrating at least comparable sample efficiency. Larger budgets (e.g., 1M–2M samples) further narrow the remaining gap to the full-size model. Moreover, the total number of samples we use is still order of magnitude smaller compared to the original pretraining corpus of these LLMs, underscoring the practicality of PnF's recovery phase.

---

> > > > ### Author Response · Authors · 2025-11-21
> > > > **Response to Weakness 4**
> > > >
> > > > We agree that comparing against analytical low‑rank baselines is essential for demonstrating the effectiveness and novelty of PnF. Accordingly, we have added SliceGPT--a closed‑form, per‑layer slicing method--as a primary baseline in all of our experiments. Across every setting we evaluate (including Qwen‑3‑4B, Qwen‑3‑8B, and multiple compression ratios), PnF consistently achieves higher average downstream performance than SliceGPT while operating under the same parameter budget and using the identical recovery dataset. To further strengthen the analysis, we also incorporate SVD‑LLM and LLM‑Pruner as additional comparison points, and PnF continues to outperform these analytical and structured‑compression approaches. This expanded set of baselines underscores the advantage of our adapter‑only, weight‑preserving pipeline.
> > > >
> > > > **Table 1.** Performance comparison between LLM-Pruner and PnF across multiple benchmarks.
> > > >
> > > > | Method            | PIQA | HellaSwag | WinoGrande | ARC-e | ARC-c | OpenBookQA | BoolQ | AvgMetric |
> > > > |-------------------|:----:|:---------:|:----------:|:-----:|:-----:|:----------:|:-----:|:---------:|
> > > > | LLM-Pruner        | 0.76 |   0.68    |    0.65    | 0.63  | 0.38  |   0.40     | 0.70  |   0.60    |
> > > > | Ours (LLM-Pruner) | 0.78 |   0.70    |    0.69    | 0.73  | 0.41  |   0.42     | 0.75  |   0.62    |
> > > >
> > > > **Table 2.** Performance comparison between PnF with SVD-LLM across multiple benchmarks.
> > > >
> > > > | Method  | PIQA | HellaSwag | WinoGrande | ARC-e | OpenBookQA | GSM8K | MathQA | TruthfulQA | AvgMetric |
> > > > |--------|:----:|:---------:|:----------:|:-----:|:----------:|:-----:|:------:|:----------:|:---------:|
> > > > | SVD-LLM| 0.69 |   0.52    |    0.68    | 0.59  |    0.33    | 0.08  |  0.26  |    0.28    |   0.43    |
> > > > | Ours   | 0.76 |   0.56    |    0.69    | 0.73  |    0.33    | 0.09  |  0.26  |    0.38    |   0.48    |

---

> ### Author Response · Authors · 2025-11-21
> **Response to Question 1**
>
> We thank the reviewer for this question and for pointing out the ambiguity in our wording. In the paper, “weight-preserving” is intended to describe the training procedure, not a formal guarantee about representational similarity. Concretely, in PnF, the pretrained backbone weights $W$ are frozen throughout training and only adapter parameters P are updated; the final compressed weights are obtained as $W^{\text{comp}} = W P$. Thus, “weight-preserving” means that the original pretrained weights are never directly modified, in contrast to pruning or low-rank editing methods that explicitly alter $W$.
>
> We do not claim that the folded model must retain identical internal representations (e.g., in terms of CKA or Jacobian similarity) to the original model—since PnF changes effective ranks and layer shapes, some representation shift is expected. Instead, our empirical evidence targets functional preservation.
>
> We compare two pipeline under the same compression rank (20\%) and training budget (600K), while disentangling **folding after adaptation** from **training after folding**.
>
> - **PnF (adapt-before-folding)**: The pretrained backbone is frozen; only lightweight adapters are trained with KL‑distillation and subsequently folded into a dense compressed matrix (our default pipeline).
> - **Baseline (train‑after‑folding)**: The model is first reduced to the same rank‑r (i.e., $r_\text{head}$, $r_\text{inter}$) using a projector, and the resulting low‑rank weights are fine‑tuned directly without any adapters.
>
> **Table 1.** Comparison of the Baseline (train-after-folding) and PnF (adapt-before-folding) configurations.
>
> | Method   | Comp.% | Sample |  PIQA |   HS  |   WG  | CSQA | ARC-e | ARC-c | OBQA | BoolQ | SIQA | MMLU |  LD  | AvgMetric |
> |----------|:------:|:------:|:-----:|:-----:|:-----:|:----:|:-----:|:-----:|:----:|:-----:|:----:|:----:|:----:|:---------:|
> | Baseline |  20%   | 600k   | 0.704 | 0.636 | 0.642 | 0.740| 0.675 | 0.416 | 0.334| 0.772 | 0.473| 0.607| 0.621|  0.602    |
> | PnF      |  20%   | 600k   | 0.736 | 0.662 | 0.669 | 0.779| 0.704 | 0.435 | 0.382| 0.783 | 0.501| 0.657| 0.651|  0.632    |
>
> Under the same rank and training budget, PnF consistently achieves better downstream performance, indicating that the frozen-W, adapter-based parameterization is more effective for preserving pretrained behavior than jointly training W and P in this setting. These results are reported in Table 1 and will be described explicitly in the revised manuscript.
>
> To clarify the intuition behind “weight-preserving,” we will also add a small 2×3 toy example in the appendix: naive slicing/truncation can be seen as applying a fixed projector that discards an entire column of W, so part of the pretrained weight matrix can never influence the output, whereas PnF learns a projector P such that all entries of W remain inside W P and can still contribute to the compressed outputs.
>
> **Backbone**
>
> $
> \begin{bmatrix}
> a \\\\
> b
> \end{bmatrix}  ^ \top
> \begin{bmatrix}
> c & d & e \\\\
> f & g & h
> \end{bmatrix}
> = \begin{bmatrix}
> ac + bf & ad + bg & ae + bh
> \end{bmatrix}
> $
>
>
> **Truncation-first**
>
> $
> \begin{bmatrix}
> a \\\\
> b
> \end{bmatrix}  ^ \top
> \begin{bmatrix}
> c & d \\\\
> f & g
> \end{bmatrix}
> = \begin{bmatrix}
> ac + bf & ad + bg
> \end{bmatrix}
> $
>
> **Adapt-befor-folding (Ours)**
>
> $
> \begin{bmatrix}
> a \\\\
> b
> \end{bmatrix}  ^ \top
> \begin{bmatrix}
> c & d & e \\\\
> f & g & h
> \end{bmatrix}
> \begin{bmatrix}
> i & j \\\\
> k & l \\\\
> m & n
> \end{bmatrix}
> = \begin{bmatrix}
> (ac + bf)i + (ad + bg)k + (ae + bh)m & (ac + bf)j + (ad + bg)l + (ae + bh)n
> \end{bmatrix}
> $
>
> We will clearly present this as informal motivation rather than a formal representational-similarity guarantee, and we will revise our terminology to phrases such as “weight-preserving training via frozen backbone + learnable adapters” to avoid misunderstanding.

---

> ### Author Response · Authors · 2025-11-21
> **Response to Question 2**
>
> We thank the reviewer for the suggestion to examine low‑bit quantization. To evaluate compatibility, we applied the same 4‑bit quantization pipeline to both the original Qwen‑3‑4B‑Base model and its PnF‑compressed versions at 20 %, 30 %, and 40 % compression ratios. Across these settings the performance degradation after quantization is not larger for the PnF models; in many cases the drop is actually smaller than for the uncompressed baseline. This indicates that the folded weights produced by PnF do not introduce harmful outliers that would be amplified by aggressive quantization. We will add these 4‑bit quantization results to the revised manuscript as further empirical evidence that PnF remains fully compatible with standard low‑bit quantization techniques.
>
> |      Method     | Comp. Ratio |   PIQA  | HellaSwag | WinoGrande |  CSQA  |  ARC-e  |   ARC-c  | openbookqa |  boolq |  siqa  |   mmlu  | lambada_openai |   Avg  | Drop rate |
> |:---------------:|:-----------:|:-------:|:---------:|:----------:|:------:|:-------:|:--------:|:----------:|:------:|:------:|:-------:|:--------------:|:------:|:---------:|
> |      Metric     |      -      | acc_nom |  acc_nom  |     acc    |   acc  | acc_nom | acc_norm |  acc_norm  |   acc  |   acc  |   acc   |       acc      |        |           |
> | Baseline        | 0% (4.02B)  | 0.7786  | 0.7364    | 0.7032     | 0.8272 | 0.7597  | 0.5162   | 0.4120     | 0.8299 | 0.5015 | 0.7131  | 0.6898         | 0.6789 | 0         |
> | Baseline (4bit) | 0% (4.02B)  | 0.7758  | 0.7200    | 0.6811     | 0.7721 | 0.7214  | 0.4906   | 0.3940     | 0.8086 | 0.4918 | 0.6774  | 0.5525         | 0.6441 | 5.13% (↓) |
> |                 |             |         |           |            |        |         |          |            |        |        |         |                |        |           |
> | | 20% (3.22B) | 0.7363  | 0.6622    | 0.6690     | 0.7790 | 0.7044  | 0.4358   | 0.3820     | 0.7837 | 0.5013 | 0.6573  | 0.6514         | 0.6329 | 0         |
> | Ours            | 30% (2.82B) | 0.7118  | 0.5878    | 0.6177     | 0.6275 | 0.6649  | 0.3803   | 0.3620     | 0.7485 | 0.4639 | 0.5243  | 0.5951         | 0.5713 | 0         |
> | | 40% (2.41B) | 0.7015  | 0.5134    | 0.5874     | 0.4195 | 0.5517  | 0.3095   | 0.3420     | 0.6846 | 0.4210 | 0.3949  | 0.5423         | 0.4971 | 0         |
> |                 |             |         |           |            |        |         |          |            |        |        |         |                |        |           |
> |     | 20% (3.22B) | 0.7341  | 0.6436    | 0.6611     | 0.7424 | 0.6704  |  0.4221  | 0.3660     | 0.7632 | 0.4839 | 0.6209  | 0.5549         | 0.6244 | 1.13% (↓) |
> | Ours (4bit)     | 30% (2.82B) | 0.7142  | 0.5698    | 0.6091     | 0.5759 | 0.6489  | 0.3760   | 0.3520     | 0.7054 | 0.4481 | 0.5002  | 0.5000         | 0.5454 | 4.53% (↓) |
> |      | 40% (2.41B  | 0.6874  | 0.4985    | 0.5743     | 0.4021 | 0.5349  | 0.2969   | 0.3160     | 0.6547 | 0.4063 | 0.3765  | 0.4439         | 0.4720 | 5.05% (↓) |

---

### Official Review · Reviewer_mpyM · 2025-10-27

**Soundness:** 3
**Presentation:** 3
**Contribution:** 2
**Rating:** 4
**Confidence:** 4

**Summary:**

This paper introduces Plug-and-Fold (PnF), a structured compression method for Large Language Models (LLMs) that preserves the original pre-trained weights and model architecture. Instead of removing components like neurons or layers, PnF inserts lightweight, learnable adapter modules into the projection layers of the attention and feed-forward networks. During a multi-stage training process, only these adapters are trained while the original weights remain frozen. After training, the adapters are folded back into the base weights via simple matrix multiplication, resulting in a compressed model that is structurally identical to the original and incurs no inference overhead. Extensive experiments across various model scales and compression rates demonstrate that PnF consistently outperforms state-of-the-art structured pruning baselines, particularly on knowledge-intensive tasks, highlighting the benefit of retaining the integrity of pre-trained weights.

**Strengths:**

1.The core innovation is a weight-preserving approach. By keeping pre-trained weights frozen and only training small adapters, the method retains the knowledge and expressive power of the original model, leading to better performance recovery after compression.

2.The method is rigorously evaluated against strong baselines on multiple models and benchmarks. It demonstrates consistent and often significant performance improvements, especially at higher compression rates, establishing a new state-of-the-art for structured compression.

3.The proposed training pipeline—comprising compression planning, group-wise sequential training, and KL-divergence distillation—is well-designed to stabilize training, prevent covariate shift, and effectively align the compressed model's output with the original model's.

**Weaknesses:**

1.While group-wise sequential training enhances stability, it also introduces a more complex and potentially longer training procedure compared to normal fine-tuning methods.

2.The use of varying compression rates across different layers results in an irregular model structure, which can introduce challenges and inconvenience for practical deployment.

3.It should be noted that the approach of compressing original weight matrices through adapter modules is not novel in the literature. Previous works such as WID[1] and SP3[2] have explored similar methodologies, where trainable compression matrices are inserted before and/or after the original weight matrices. These compression matrices are subsequently merged back into the original weights through matrix multiplication after training to achieve the final compressed form. However, the authors did not cite these two relevant papers in their work.

[1]Wu, Taiqiang, et al. "Weight-inherited distillation for task-agnostic bert compression." arXiv preprint arXiv:2305.09098 (2023).

[2]Hu, Yuxuan, et al. "$\rm SP^ 3$: Enhancing Structured Pruning via PCA Projection." arXiv preprint arXiv:2308.16475 (2023).

**Questions:**

Did the author know about the two highly similar papers, WID and SP3? The method proposed in this paper compresses by directly introducing a low-dimensional compression matrix, whereas WID and SP3 introduce higher-dimensional compression matrices and compress them during training. In my opinion, there is not much fundamental difference between these two approaches.

---

> ### Author Response · Authors · 2025-11-21
> **Response to Weakness 1**
>
> We appreciate the reviewers’ thoughtful comments and the opportunity to clarify these points.
>
>
>
> We acknowledge that group-wise sequential training introduces additional complexity of the training pipeline and can increase overall training time compared to a single-stage procedure. To make this trade-off explicit, table1 summarizes training time and average downstream zero-shot performance for Qwen3-4B (8×H100) under the same 20% compression setting. As shown in table, non-sequential (all-at-once) PnF uses a single-stage training loop similar in structure to standard fine-tuning, yet it already recovers strong performance surpasses the baselines. By contrast, group-wise sequential training increases recovery phase by roughly a factor of three, but yields a clear additional gain in average performance (about +3% absolute).
>
> We will emphasize that the sequential schedule is optional rather than a requirement of PnF. Practitioners with limited wall-clock budget can use the single-stage variant, whereas or those who prioritize maximum accuracy and stability can switch to the multi-stage variant. This flexibility allows users to choose the configuration that best matches their resource constraints and performance goals.
>
> **Table 1.** Training latency and average performance for the two training regimes.
>
> | Regime              | # Groups | Training Latency | Performance (avg.) |
> |-----------------------|:--------:|------------------|--------------------|
> | Non-sequential Training |    1     | 8.21 hours       | 0.602              |
> | Sequential Training     |    4     | 23:32 hours      | 0.633              |

---

> > ### Author Response · Authors · 2025-11-21
> > **Response to Weakness 2**
> >
> > We appreciate the reviewer’s emphasis on deployability, and we would like to clarify that PnF was explicitly designed for practical deployment. First, after training stage, all PnF adapters are folded into the backbone weights, so the deployed model has the same number of layers, block structure, and external hidden dimension as the original Transformer. The resulting model is still a static graph of standard dense matrix multiplications, where each linear layer is a GEMM with its own (m, k, n) dimensions, as in the standard BLAS-style interface.
> >
> > Second, non-uniform capacity across layers is already a standard architectural design. The MatFormer [1] architecture assigns different FFN widths to different layers to enable elastic inference, and this design is deployed in Google's Gemma 3n models [2] specifically to support efficient serving under diverse latency and cost constraints.
> >
> > Third, the pruning works provides evidence that uniform pruning ratios are sub-optimal. [3] and [4] both adopt adaptive, layer-wise (or module-wise) pruning rates under a global parameter budget and report better accuracy-efficiency trade-offs than uniform sparsity schemes, indicating that non-uniform compression across layers is not only practical but often preferable in modern LLM compression.
> >
> > To make these points explicit, we will revise the manuscript to emphasize that (1) the deployed PnF model preserves the original Transformer architecture and uses still standard dense GEMM operation, and (2) our layer-wise non-uniform compression follows prevailing practices in recent LLM architectures (e.g., MatFormer / Gemma 3n) and pruning method with adaptive pruning ratio allocation.
> >
> > [1] Devvrit, F. et al., "MatFormer: Nested Transformer for Elastic Infernce", NeurIPS 2024.
> >
> > [2] Google DeepMind, "Gemma 3n model overview", Google AI for Developers, 2025.
> >
> > [3] Ban, Z. et al., "GAP: A Global Adaptive Pruning Method for Large Language Models, EMNLP 2025.
> >
> > [4] Yang, M. et al., "Let LLM Tell What do Prune and How Much to Prune", ICML 2025.

---

> ### Author Response · Authors · 2025-11-21
> **Response to Weakness 3**
>
> We appreciate the reviewer’s careful reading of our manuscript and for bringing the closely related works WID and SP$^3$ to our attention. We acknowledge that these papers share the high‑level idea of using adapter-based parameterization that are later folded into the backbone, and we regret that they were not cited in the current version.
>
>
> Briefly, WID [1] introduce trainable row/column "compactors" around linear layers in BERT, update both compactors and original weights, and then uses learned row/column masks (based on small-norm directions) to prune the compactors before folding them into the backbone. SP$^3$ [2] uses PCA-based projection matrices together with continuous mask variables to reduce hidden dimensions, and then fine-tunes the pruned backbone as a standard trainable model.
>
>
> PnF is indeed in this adapter-as-compression-and-then-folding family, but **differs in several key aspects**:
>
> - **Frozen backbone, adapter-only training**: We keep all pretrained weight $W$ fixed and update **only low-rank adapters $P$**, forming compressed weights as  $W^{Comp}=WP$.  By contrast, both WID and SP$^3$ jointly update the original weights and the re-parameterized components, modifying the backbone parameters during optimization.
>
> - **No learned pruning masks**: PnF does not learn or apply any row/column/channel masks; each layer remains a dense matrix before and after folding, whereas WID and $SP^3$ both rely on learned masks to define which directions are pruned.
>
> - **Multi-stage weight-preserving structural plan for decoder LLMs**: PnF is designed for decoder only LLMs (OPT, Qwen, LLaMA-3) and applies group-wise sequential training with pyramid compression planing in which trains adapters on a frozen backbone and then fold them. Our experiments show that this purely adapter-based, weight-preserving pipeline successfully recovers performance across multiple compression stages at multi-billion-parameter scale, which is different in spirit from the mask-based, jointly-trained pruning procedures used in WID and SP$^3$.
>
> We will revise the manuscript to cite WID [1] and SP$^3$ [2], acknowledge them as prior instances of adpater-style parameterization for compression, and clearly state how PnF differs along these axes.

---

> > ### Comment · Reviewer_mpyM · 2025-11-21
> >
> > Thanks for the authors' response and clarification. I find this work to be meaningful, as it extends previous weight-inheritance-based methods from BERT to LLMs and achieves promising performance.
> >
> > The primary concern I had with PnF is the higher time cost of its group-wise sequential training compared to standard fine-tuning. I would suggest that it might be better to frame standard fine-tuning as the primary method, with group-wise sequential training presented as an extension for scenarios with a sufficient computational budget.
> >
> > Additionally, the authors should acknowledge the contributions of prior methods (WID and SP3) in the revised manuscript and clarify the distinctions between PnF and these approaches.
> >
> > Thank you again for the detailed response. I have raised my score to support the acceptance of this paper.

---

> > > ### Author Response · Authors · 2025-11-24
> > >
> > > Thank you for your thoughtful feedback. We appreciate your kind comments and will incorporate the points you raised into the revised manuscript to further improve its completeness. Once again, thank you for the supportive review and for raising the score.

---

### Official Review · Reviewer_4ad1 · 2025-11-01

**Soundness:** 3
**Presentation:** 3
**Contribution:** 2
**Rating:** 4
**Confidence:** 3

**Summary:**

Plug-and-Fold (PnF) is a new structured compression framework for large language models (LLMs). It keeps the pretrained backbone intact while inserting small, trainable adapters into attention and feed-forward layers. Once trained with frozen base weights, the adapters are folded into the model through simple matrix multiplications, producing a compressed variant structurally identical to the original and free of runtime overhead. We validate PnF on multiple LLMs, including Qwen and OPT families.

**Strengths:**

1. The paper is clearly organized, and the figures effectively illustrate the folding pipeline.

2. The paper includes group-wise and data-size ablations that show trade-offs among compression rate, accuracy, and recovery set size, providing useful insights into design choices.

**Weaknesses:**

1. PnF is conceptually close to AdaLoRA [4], which also trains low-rank adapters and merges them into base weights, but differs mainly in emphasizing structural identity after folding. The analogy to LLM-Pruner [5] is less precise—PnF does not perform pruning but instead adds and folds adapters. Overall, the method introduces limited theoretical or algorithmic innovation beyond existing adapter-based compression frameworks.

2. Evaluations partially miss recent baselines. While SliceGPT [1], LaCo [2], and ShortGPT [3] are included, stronger or newer methods such as LLM-Pruner [5] and DISP-LLM [6] are absent under matched compression ratios and datasets. Their inclusion would strengthen the empirical comparison.

3. The paper claims “deployment efficiency” but provides no FLOP, latency, or energy analyses. Such quantitative metrics are essential to substantiate statements like “no runtime overhead” and to contextualize benefits relative to SliceGPT [1] or LLM-Pruner [5].

4. Experiments cover only the OPT and Qwen series, which limits generality. Additional results on diverse architectures, such as LLaMA-3, T5, or Mixtral, would better demonstrate robustness and broader applicability.

4. The manual grouping schedule is empirical and could be automated. The paper itself acknowledges that this stage depends on practitioner heuristics. Replacing it with learned or adaptive strategies, such as sensitivity-based or reinforcement-learning layer selection as in DISP-LLM [6], would enhance reproducibility.

5. The “weight-preserving” claim is supported only by performance metrics. Analyses of representational similarity (e.g., centered kernel alignment (CKA) [8]) or Jacobian similarity would substantiate that the folded model genuinely retains the representational structure of the original weights.

6. Section 3.2 would benefit from concise pseudocode summarizing the plug-and-fold pipeline. Presenting the compression planning, group-wise training, and folding steps in algorithmic form would improve reproducibility and clarify the workflow.

[1] S. Ashkboos et al. SliceGPT: Compress Large Language Models by Deleting Rows and Columns. 2024.

[2] Y. Yang et al. LaCo: Large Language Model Pruning via Layer Collapse. 2024.

[3] X. Men et al. ShortGPT: Layers in Large Language Models Are More Redundant Than You Expect. 2024.

[4] Q. Zhang et al. AdaLoRA: Adaptive Budget Allocation for Parameter-Efficient Fine-Tuning. 2023.

[5] X. Ma et al. LLM-Pruner: On the Structural Pruning of Large Language Models. 2023.

[6] S. Gao et al. DISP-LLM: Dimension-Independent Structural Pruning for Large Language Models. 2024.

[7] A. Gromov et al. The Unreasonable Ineffectiveness of the Deeper Layers. 2024.

[8] S. Kornblith et al. Similarity of Neural Network Representations Revisited. 2019.

**Questions:**

1. How does PnF differ algorithmically from existing merging or folding approaches such as AdaLoRA [4] and LLM-Pruner [5], which also integrate low-rank or pruned components into base weights?

2. Could the authors include evaluations against recent structured compression baselines, such as SliceGPT [1], LaCo [2], ShortGPT [3], LLM-Pruner [5], and DISP-LLM [6], under identical compression ratios and datasets for fair comparison?

3. What are the quantitative FLOP, latency, and energy reductions of PnF compared with SliceGPT [1] and LLM-Pruner [5], and how do these relate to the claimed deployment efficiency?

4. Have the authors tested PnF on architectures beyond OPT and Qwen (e.g., LLaMA-3, T5, Mixtral) to demonstrate cross-architecture robustness and folding consistency?

5. Could the authors consider replacing the manual grouping plan with an adaptive or automated selection strategy?

---

> ### Author Response · Authors · 2025-11-21
> **Response to Weakness 1**
>
> We appreciate Reviewer 4ad1’s insightful suggestions for strengthening the arguments presented in our paper. We will incorporate your suggestions as outlined in the review:
>
>
>
> While PnF indeed uses low-rank adapters, we would like to clarify two key conceptual and algorithmic differences.
>
> **(1) PnF is a structural compression method, not a PEFT method like AdaLoRA**
>
> AdaLoRA [4] is designed for parameter-efficient fine-tuning: it learns low-rank increments $\Delta W$ on top of a fixed backbone, adaptively allocates LoRA rank across layers, and finally merges $\Delta W$ into $W$, but **the model architecture (hidden size, head dimensions, FFN widths) and parameter count remain unchanged** at inference. In contrast, PnF treats **low-rank adapters as a tool for changing the backbone structure itself**: the adapters are constrained to be **foldable into smaller weight matices (e.g., reduced head dimension and intermediate size)**, and after folding we obtain a compressed backbone with fewer parameters and strictly smaller layer shapes. Moreover, AdaLoRA optimizes task-specific losses on downstream datasets, whereas PnF uses **task-agnostic knowledge distillation over a broad mixture to preserve the original model's general behavior**. Therefore, although both methods employ low-rank adapters, their goals (PEFT vs. model compression), constraint (no shape change vs. enforced shape reduction), and training objectives are fundamentally different. Highlighting these similarities, however, will undoubtedly improve the readability of our paper and will be applied to our revised version.
>
> **(2) Relation to LLM-Pruner is conceptual (capacity allocation), not algorithmic pruning**
>
> We acknowledge that PnF does not perform pruning in the strict sense; unlike LLM‑Pruner [5], which removes heads or channels based on gradient‑derived importance scores, PnF never deletes parameters. Our reference to LLM‑Pruner is therefore meant as a conceptual link: both methods aim at task‑agnostic compression of large language models and both stress non‑uniform, layer‑wise capacity allocation under a global budget—for example, compressing later layers more aggressively than earlier ones. **PnF realizes this idea in a different manner**. Instead of pruning individual components, we devise a **pyramidal structural compression plan that determines a hierarchy of reduced layer sizes**.
>
> To address the reviewer’s concern, we will clarify that PnF’s main contribution lies in framing weight‑preserving structured compression as a foldable low‑rank‑adapter pipeline for genuine architecture reduction—not merely as a PEFT technique. In the related‑work discussion we will more clearly separate the conceptual inspiration drawn from LLM‑Pruner’s task‑agnostic capacity allocation from the concrete algorithmic components that are unique to PnF.

---

> ### Author Response · Authors · 2025-11-21
> **Response to Weakness 2**
>
> We agree that including LLM-Pruner and DISP-LLM as baselines would make the empirical comparison more complete. Therefore, we have conducted experiments to compare PnF with the suggested methods.
>
> **(1) LLM-Pruner [5]**
> We align with the setting used in LLM-Pruner by adopting LLaMA-2-7B and comparing at a 20\% compression ratio on the shared benchmarks (PIQA, HellaSwag, WinoGrande, ARC-e, ARC-c, OpenbookQA, and BoolQ). We use the results reported in [5] and place them alongside our PnF results in a new table. Under this setting, PnF attains a higher average performance than LLM-Pruner over all benchmarks.
>
> **Table 1.** Benchmark comparison between LLM-Pruner and PnF.
>
> | Method            | PIQA | HellaSwag | WinoGrande | ARC-e | ARC-c | OpenBookQA | BoolQ | AvgMetric |
> |-------------------|:----:|:---------:|:----------:|:-----:|:-----:|:----------:|:-----:|:---------:|
> | LLM-Pruner        | 0.76 |   0.68    |    0.65    | 0.63  | 0.38  |   0.40     | 0.70  |   0.60    |
> | Ours (LLM-Pruner) | 0.78 |   0.70    |    0.69    | 0.73  | 0.41  |   0.42     | 0.75  |   0.62    |
>
> **(2) DISP-LLM [6]**
> Similarly, DISP-LLM is added as another recent structured compression baseline. Using LLaMA-2-7B and the same 20\% compression setting, we compare our PnF results to the DISP-LLM performance reported in [6] on the reported benchmarks. PnF again achieves higher performance across all benchmarks.
>
> **Table 2.** Benchmark comparison between DISP-LLM and PnF. “--” denotes a missing value.
>
> | Method           | PIQA | HellaSwag | WinoGrande | ARC-e | ARC-c | OpenBookQA | BoolQ | AvgMetric |
> |------------------|:----:|:---------:|:----------:|:-----:|:-----:|:----------:|:-----:|:---------:|
> | DISP-LLM         | 0.77 |   0.68    |    0.65    | 0.65  | 0.37  |    --      |  --   |   0.62    |
> | Ours (DISP-LLM)  | 0.78 |   0.70    |    0.69    | 0.73  | 0.41  |    --      |  --   |   0.66    |
>
>
> These two baselines and the accompanying comparison table will be added to the revised manuscript to strengthen the empirical evidence for PnF.

---

> ### Author Response · Authors · 2025-11-21
> **Response to Weakness 3**
>
> We thank the reviewer for highlighting the potential ambiguity of the term **deployment efficiency**. Our original intent was to convey that, after compression, the PnF model can be deployed without any modifications to the inference stack. i.e. it retains the same static Transformer graph and relies on standard dense GEMM operators as the original backbone. No custom sparse kernels, code generation, or additional runtime components are required. To prevent misinterpretation, we will replace the original wording with a more precise description: "deployment-friendly: no custom kernel, standard dense operators". We believe these revisions eliminate any possible confusion and provide precise description of PnF’s practical advantage during inference.
>
> We also agree that quantitative metrics are helpful. We report average per-token inference time for original backbone, PnF model, and  SliceGPT under various compression ratios (i.e., 20\%, 30\%, and 40\%), the same backbone and hardware (i.e., Qwen3-4B-Base and 1*H100). Our results show that the average per-token inference time consistently decreases as we increase the compression ratio to 20\%, 30\%, and 40\%, and that at the same compression ratio.
>
> **Table 1.** Latency per-token generation of SliceGPT vs. our method at different compression ratios.
>
> | Method   | 0% (No comp) | 20%      | 30%      | 40%      |
> |----------|--------------|----------|----------|----------|
> | SliceGPT | 29.126 ms    | 22.299 ms| 22.120 ms| 22.099 ms|
> | Ours     | 29.126 ms    | 22.081 ms| 21.719 ms| 21.559 ms|

---

> ### Author Response · Authors · 2025-11-21
> **Response to Weakness 4**
>
> We agree that evaluating across multiple model families is important for assessing generality. In addition to OPT and Qwen, we have additionally evaluate PnF on LLaMA-3.1-8B under the same experimental protocol and compression ratios of 20\%, 30\%, and 40\%. Moreover, we are currently training PnF on LLaMA-3.2-3B and report the those results once the experiments complete. In case of compression Llama-3.1-8B, across all three compression ratios, PnF achieves the best average performance among the structured compression baselines we consider (e.g., SliceGPT, LaCO, LLM-Streamline, and ShortGPT) on the shared benchmarks. This confirms that the same trend observed on OPT and Qwen also extend to the LLaMA-3-style architecture. We will incorporate these LLaMA-3.1-8B results and their summary table into the revised manuscript as well.

---

> ### Author Response · Authors · 2025-11-21
> **Experimental Results 1 for the Response to Weakness 4**
>
> **Table 1.**  Zero-shot performance of the various compression methods on Llama-3.1-8B. The pretrained backbone model and its compressed variants are evaluated across multiple benchmarks at several compression rates
>
> | Method                      | PIQA   | HellaSwag | WinoGrande | CSQA   | ARC-e  | ARC-c  | openbookqa | boolq  | siqa   | mmlu   | lambada_openai | Avg    |
> |-----------------------------|--------|-----------|------------|--------|--------|--------|------------|--------|--------|--------|----------------|--------|
> |                             |        |           |            |        |        |        |            |        |        |        |                |        |
> | Baseline 0% (8.03B)         | 0.8123 | 0.7884    | 0.7356     | 0.715  | 0.8123 | 0.5367 | 0.446      | 0.8196 | 0.4713 | 0.6345 | 0.7533         | 0.6841 |
> | Slice GPT ~20% (6.41B)      | 0.5582 | 0.3818    | 0.5414     | 0.2015 | 0.3068 | 0.2449 | 0.284      | 0.5535 | 0.3511 | 0.2466 | 0.0726         | 0.3402 |
> | Slice GPT ~30% (5.61B)      | 0.5854 | 0.3592    | 0.5335     | 0.1966 | 0.3603 | 0.2568 | 0.276      | 0.459  | 0.3403 | 0.2376 | 0.0819         | 0.3351 |
> | Slice GPT ~40% (4.83B)      | 0.5609 | 0.3299    | 0.5067     | 0.1957 | 0.3439 | 0.2321 | 0.256      | 0.4367 | 0.3454 | 0.2461 | 0.077          | 0.321  |
> | LaCo ~20% (6.50B)           | 0.7693 | 0.7056    | 0.6875     | 0.5209 | 0.7155 | 0.4317 | 0.38       | 0.7691 | 0.4565 | 0.4671 | 0.6534         | 0.5961 |
> | LaCo ~30% (5.63B)           | 0.728  | 0.6209    | 0.663      | 0.3604 | 0.6233 | 0.3558 | 0.35       | 0.6667 | 0.435  | 0.3478 | 0.5694         | 0.52   |
> | LaCo ~40% (4.76B)           | 0.667  | 0.5141    | 0.6243     | 0.421  | 0.5139 | 0.3055 | 0.29       | 0.6312 | 0.4181 | 0.4141 | 0.4809         | 0.48   |
> | LLM-Streamline ~20% (6.50B) | 0.7514 | 0.7007    | 0.7238     | 0.6912 | 0.7214 | 0.4633 | 0.394      | 0.7609 | 0.4585 | 0.6164 | 0.3872         | 0.6062 |
> | LLM-Streamline ~30% (5.63B) | 0.6986 | 0.6035    | 0.6906     | 0.7035 | 0.617  | 0.3763 | 0.362      | 0.7593 | 0.436  | 0.6271 | 0.3949         | 0.5699 |
> | LLM-Streamline ~40% (4.76B) | 0.6785 | 0.4778    | 0.5872     | 0.1941 | 0.5059 | 0.2833 | 0.326      | 0.619  | 0.4007 | 0.2301 | 0.275          | 0.4161 |
> | Short GPT ~20% (6.50B)      | 0.7465 | 0.6924    | 0.7159     | 0.6986 | 0.7024 | 0.4437 | 0.374      | 0.7214 | 0.4611 | 0.5919 | 0.7075         | 0.6232 |
> | Short GPT ~30% (5.63B)      | 0.6855 | 0.5914    | 0.6993     | 0.5872 | 0.5875 | 0.3626 | 0.306      | 0.7113 | 0.4222 | 0.4208 | 0.541          | 0.5377 |
> | Short GPT ~40% (4.76B)      | 0.6213 | 0.4531    | 0.5983     | 0.1974 | 0.431  | 0.2807 | 0.278      | 0.6226 | 0.3889 | 0.2302 | 0.3427         | 0.404  |
> | Ours ~20% (6.42B)           | 0.775  | 0.7369    | 0.713      | 0.6326 | 0.7494 | 0.4654 | 0.404      | 0.7942 | 0.4606 | 0.617  | 0.7226         | 0.6428 |
> | Ours ~30% (5.62B)           | 0.7584 | 0.7048    | 0.6932     | 0.5892 | 0.7149 | 0.4389 | 0.392      | 0.7525 | 0.4503 | 0.5713 | 0.6942         | 0.6145 |
> | Ours ~40% (4.48B)           | 0.7439 | 0.624     | 0.6438     | 0.4562 | 0.6135 | 0.3494 | 0.362      | 0.7094 | 0.4368 | 0.4528 | 0.6219         | 0.5467 |

---

> ### Author Response · Authors · 2025-11-21
> **Experimental Results 2 for the Response to Weakness 4**
>
> **Table 2.**  Five-shot performance of the various compression methods on Llama-3.1-8B. The pretrained backbone model and its compressed variants are evaluated across multiple benchmarks at several compression rates
>
> | Method                      | PIQA   | HellaSwag | WinoGrande | CSQA   | ARC-e  | ARC-c  | openbookqa | boolq  | siqa   | mmlu   | lambada_openai | Avg    |
> |-----------------------------|--------|-----------|------------|--------|--------|--------|------------|--------|--------|--------|----------------|--------|
> |                             |        |           |            |        |        |        |            |        |        |        |                |        |
> | Baseline 0% (8.03B)         | 0.8243 | 0.8092    | 0.7719     | 0.7412 | 0.8502 | 0.5768 | 0.464      | 0.8275 | 0.5251 | 0.6503 | 0.6848         | 0.7023 |
> | Slice GPT ~20% (6.41B)      | 0.5756 | 0.4066    | 0.5541     | 0.1916 | 0.3359 | 0.2491 | 0.302      | 0.6355 | 0.3675 | 0.2339 | 0.066          | 0.3562 |
> | Slice GPT ~30% (5.61B)      | 0.6023 | 0.3787    | 0.532      | 0.199  | 0.4398 | 0.2773 | 0.252      | 0.541  | 0.3593 | 0.2385 | 0.0681         | 0.3535 |
> | Slice GPT ~40% (4.83B)      | 0.574  | 0.335     | 0.5217     | 0.2113 | 0.3826 | 0.2432 | 0.262      | 0.433  | 0.3582 | 0.2465 | 0.053          | 0.3291 |
> | LaCo ~20% (6.50B)           | 0.7688 | 0.7243    | 0.6953     | 0.5356 | 0.7437 | 0.4539 | 0.386      | 0.7639 | 0.4985 | 0.449  | 0.614          | 0.603  |
> | LaCo ~30% (5.63B)           | 0.7274 | 0.6413    | 0.6630      | 0.4357 | 0.681  | 0.3891 | 0.356      | 0.7131 | 0.4724 | 0.4323 | 0.483          | 0.5449 |
> | LaCo ~40% (4.76B)           | 0.6654 | 0.5273    | 0.6243     | 0.4595 | 0.5581 | 0.3038 | 0.312      | 0.6636 | 0.4437 | 0.4385 | 0.4407         | 0.4943 |
> | LLM-Streamline ~20% (6.50B) | 0.7563 | 0.7269    | 0.7648     | 0.7314 | 0.7626 | 0.4804 | 0.408      | 0.808  | 0.5164 | 0.6339 | 0.3553         | 0.6313 |
> | LLM-Streamline ~30% (5.63B) | 0.6828 | 0.5475    | 0.6709     | 0.412  | 0.5829 | 0.3473 | 0.322      | 0.6755 | 0.4478 | 0.419  | 0.3546         | 0.4966 |
> | LLM-Streamline ~40% (4.76B) | 0.6545 | 0.387     | 0.5178     | 0.2097 | 0.4718 | 0.2423 | 0.28       | 0.541  | 0.3639 | 0.2465 | 0.1679         | 0.3711 |
> | Short GPT ~20% (6.50B)      | 0.7508 | 0.7202    | 0.7388     | 0.7281 | 0.7462 | 0.4676 | 0.372      | 0.6596 | 0.5118 | 0.6318 | 0.6231         | 0.6318 |
> | Short GPT ~30% (5.63B)      | 0.6942 | 0.6085    | 0.723      | 0.6298 | 0.6237 | 0.3737 | 0.324      | 0.7095 | 0.4821 | 0.4828 | 0.53           | 0.5619 |
> | Short GPT ~40% (4.76B)      | 0.6328 | 0.4622    | 0.6212     | 0.2031 | 0.4735 | 0.2935 | 0.286      | 0.6214 | 0.4222 | 0.2793 | 0.3113         | 0.4188 |
> | Ours ~20% (6.42B)           | 0.8094 | 0.7548    | 0.7482     | 0.6714 | 0.799  | 0.5132 | 0.43       | 0.8046 | 0.5131 | 0.6262 | 0.6641         | 0.6667 |
> | Ours ~30% (5.62B)           | 0.7844 | 0.7224    | 0.7082     | 0.6063 | 0.7662 | 0.468  | 0.414      | 0.7656 | 0.4969 | 0.5842 | 0.6462         | 0.6329 |
> | Ours ~40% (4.48B)           | 0.7527 | 0.6369    | 0.6546     | 0.4776 | 0.6818 | 0.3754 | 0.376      | 0.7332 | 0.4772 | 0.4761 | 0.5814         | 0.5657 |

---

> > ### Author Response · Authors · 2025-11-21
> > **Response to Weakness 5**
> >
> > We acknowledge that the manual grouping schedule is currently empirical and depends on practitioner heuristics, as noted in the paper. Replacing this stage with a learned or adaptive strategy—such as sensitivity‑based selection or reinforcement‑learning–driven layer grouping, as demonstrated in DISP‑LLM [6]—would indeed improve reproducibility and reduce manual tuning. In our ongoing work, we are exploring automatic grouping and rank‑planning mechanisms that learn which layers to group and compress at each stage, rather than fixing the schedule by hand. We will explicitly discuss these directions in the limitation and future‑work sections and cite DISP‑LLM as a complementary approach.

---

> ### Author Response · Authors · 2025-11-21
> **Response to Weakness 6**
>
> We thank the reviewer for this comment and for pointing out the ambiguity in our wording. In our paper, “weight-preserving” is intended to describe the **frozen pretrained weight** during training procedure, not a guarantee about representational similarity after compression. Concretely, in PnF, the pretrained backbone weights $W$ are frozen throughout training, and only the adapter parameters $P$ are updated; the final compressed weights are obtained as $W^{\text{comp}} = WP$. Thus, weight-preserving means that **the original pretrained weights are never directly modified** unlike prior pruning or low-rank editing methods that explicitly change $W$.
>
> We do not claim that the folded model must retain identical internal representations (e.g., CKA or Jacobian similarity) to the original model—indeed, PnF changes the effective ranks and layer shapes, so some representational shift is expected. Our empirical evidence instead targets functional preservation, via downstream performance. In Table 1, we compare PnF (adapt-before-folding) against a train-after-folding baseline under the same rank and training budget, and observe that PnF consistently achieves better downstream performance, indicating that this weight-preserving adapter parameterization is effective for maintaining pretrained behavior at a given compression ratio.
>
> **Table 1.** Comparison of the Baseline (train-after-folding) and PnF (adapt-before-folding) configurations.
>
> | Method   | Comp.% | Sample |  PIQA |   HS  |   WG  | CSQA | ARC-e | ARC-c | OBQA | BoolQ | SIQA | MMLU |  LD  | AvgMetric |
> |----------|:------:|:------:|:-----:|:-----:|:-----:|:----:|:-----:|:-----:|:----:|:-----:|:----:|:----:|:----:|:---------:|
> | Baseline |  20%   | 600k   | 0.704 | 0.636 | 0.642 | 0.740| 0.675 | 0.416 | 0.334| 0.772 | 0.473| 0.607| 0.621|  0.602    |
> | PnF      |  20%   | 600k   | 0.736 | 0.662 | 0.669 | 0.779| 0.704 | 0.435 | 0.382| 0.783 | 0.501| 0.657| 0.651|  0.632    |
>
> To further clarify the intuition behind “weight-preserving,” we will also provide a small 2×3 toy example. In this example, naive slicing or truncation can be seen as applying a fixed projector that discards an entire column of $W$, so part of the pretrained weight matrix can never influence the output. In contrast, PnF learns a adapter $P$ so that all entries of the original pretrained weight matrix remain present inside $WP$ and can still contribute to the compressed outputs. We will clearly present this as an informal motivation, and rely on the empirical comparison in Table 1 as the primary evidence.
>
> **Backbone**
>
> $
> \begin{bmatrix}
> a \\\\
> b
> \end{bmatrix}  ^ \top
> \begin{bmatrix}
> c & d & e \\\\
> f & g & h
> \end{bmatrix}
> = \begin{bmatrix}
> ac + bf & ad + bg & ae + bh
> \end{bmatrix}
> $
>
>
> **Truncation-first**
>
> $
> \begin{bmatrix}
> a \\\\
> b
> \end{bmatrix}  ^ \top
> \begin{bmatrix}
> c & d \\\\
> f & g
> \end{bmatrix}
> = \begin{bmatrix}
> ac + bf & ad + bg
> \end{bmatrix}
> $
>
> **Adapt-befor-folding (Ours)**
>
> $
> \begin{bmatrix}
> a \\\\
> b
> \end{bmatrix}  ^ \top
> \begin{bmatrix}
> c & d & e \\\\
> f & g & h
> \end{bmatrix}
> \begin{bmatrix}
> i & j \\\\
> k & l \\\\
> m & n
> \end{bmatrix}
> = \begin{bmatrix}
> (ac + bf)i + (ad + bg)k + (ae + bh)m & (ac + bf)j + (ad + bg)l + (ae + bh)n
> \end{bmatrix}
> $
>
>
> To avoid misunderstanding, we will revise the terminology to phrases such as “weight-preserving training via frozen backbone + learnable adapters” and explicitly clarify that we are referring to how parameters are updated during training, rather than to a formal representational-similarity guarantee.

---

> ### Author Response · Authors · 2025-11-21
> **Response to Weakness 7**
>
> We thank the reviewer for this helpful suggestion and agree that a concise algorithmic summary would improve clarity and reproducibility. In the revised manuscript, we add a short pseudo-code in Appendix that explicitly summarizes the plug-and-fold pipeline, including (i) compression planning (per-layer rank allocation and selection of layers to compress), (ii) group-wise adapter training on a frozen backbone, and (iii) the folding step that replaces each compressed layer’s weight with $W_l^{\text{Comp}} = W_l P_l$. This makes it clear that adapters are attached only to layers whose dimensions are actually reduced, and it presents the full workflow in a form that is easy to re-implement.
>
> **Algorithm 1. Plug-and-Fold (PnF) Compression Pipeline**
>
> ```text
> Input:
>   - Pretrained decoder-only LLM weights {W_l}_{l=1}^L
>   - Global compression rate rho
>   - Teacher model f_teacher
>   - Distillation dataset D
> Output:
>   - Compressed weights {W_l_comp}_{l=1}^L
>
> # [Compression planning]
> 1. Compute per-layer target ranks/widths {r_l}_{l=1}^L to match the global compression rate rho.
> 2. Define the set of layers to compress:
>      C := { l in {1,...,L} : r_l < full_dim(W_l) }.
>    (Layers with r_l equal to the original dimension are not compressed.)
> 3. Partition C into groups G_1, ..., G_K.
>    (Single-stage PnF uses K = 1 and G_1 = C.)
>
> # [Group-wise sequential training]
> 4. Define PnF adapters {P_l}_{l in C}.
>
> 5. For k = 1 to K:
>      5.1 For each layer l in G_k:
>            - Attach a PnF adapter P_l to W_l (e.g., replace x W_l by x W_l P_l) with rank r_l.
>            - Freeze W_l and mark only P_l as trainable.
>      5.2 Initialize {P_l}_{l in G_k} (e.g., near-identity initialization).
>
>      5.3 For training step t = 1 to T_k:
>            - Sample a minibatch x ~ D.
>            - Compute teacher outputs:
>                p_teacher(· | x) = f_teacher(x).
>            - Compute student outputs:
>                p_student(· | x; {W_l, P_l}).
>            - Update only {P_l}_{l in G_k} by minimizing a distillation loss,
>              e.g., KL(p_teacher || p_student).
>
>      5.4 For each layer l in G_k (folding step):
>            - W_l_comp := W_l P_l.
>            - Remove P_l from the model (keep only W_l_comp at inference).
>
> 6. Return compressed model with folded weights {W_l_comp}_{l=1}^L.

---

> ### Author Response · Authors · 2025-11-26
> **Experimental Results 3 for the Response to Weakness 4**
>
> We further report additional experiments on Llama-3.2-3B to substantiate the effectiveness of our method. These results also show that PnF consistently achieves better performance than the baselines. We will incorporate the Llama-3.2-3B results into the revised manuscript.
>
> **Table 3.**  Zero-shot performance of the various compression methods on Llama-3.2-3B. The pretrained backbone model and its compressed variants are evaluated across multiple benchmarks at several compression rates
>
> | Method                         | PIQA   | HellaSwag | WinoGrande | CSQA   | ARC-e  | ARC-c  | openbookqa | boolq  | siqa   | mmlu   | lambada_openai | Avg   |
> |--------------------------------|--------|-----------|------------|--------|--------|--------|------------|--------|--------|--------|----------------|-------|
> |                                |        |           |            |        |        |        |            |        |        |        |                |       |
> | Baseline 0% (3.21B)            | 0.7748 | 0.7370    | 0.6906     | 0.6404 | 0.7168 | 0.4582 | 0.4320     | 0.7278 | 0.4708 | 0.5396 | 0.7000         | 0.6262 |
> | Slice GPT ~20% (2.90B)         | 0.5664 | 0.3318    | 0.5217     | 0.1998 | 0.3396 | 0.2543 | 0.3020     | 0.5966 | 0.3552 | 0.2420 | 0.0714         | 0.3437 |
> | Slice GPT ~30% (2.56B)         | 0.5484 | 0.3178    | 0.4996     | 0.1966 | 0.3211 | 0.2415 | 0.2580     | 0.5841 | 0.3449 | 0.2562 | 0.0732         | 0.3310 |
> | Slice GPT ~40% (2.22B)         | 0.5424 | 0.2923    | 0.4980     | 0.1957 | 0.3026 | 0.2278 | 0.2540     | 0.4798 | 0.3444 | 0.2617 | 0.0472         | 0.3133 |
> | LaCo ~20% (2.61B)              | 0.7002 | 0.6330    | 0.6890     | 0.6183 | 0.6044 | 0.3771 | 0.3520     | 0.6697 | 0.4427 | 0.5177 | 0.6330         | 0.5670 |
> | LaCo ~30% (2.21B)              | 0.6736 | 0.5134    | 0.5864     | 0.3227 | 0.5248 | 0.3131 | 0.3200     | 0.6242 | 0.4033 | 0.3148 | 0.4801         | 0.4615 |
> | LaCo ~40% (1.90B)              | 0.6028 | 0.4156    | 0.5667     | 0.2228 | 0.3952 | 0.2491 | 0.2840     | 0.6217 | 0.3915 | 0.2652 | 0.3972         | 0.4011 |
> | LLM-Streamline ~20% (2.61B)    | 0.7138 | 0.6171    | 0.6661     | 0.6372 | 0.6103 | 0.3840 | 0.3740     | 0.7150 | 0.4401 | 0.5450 | 0.5131         | 0.5651 |
> | LLM-Streamline ~30% (2.21B)    | 0.6763 | 0.5317    | 0.6504     | 0.4390 | 0.5459 | 0.3345 | 0.3160     | 0.6450 | 0.4150 | 0.4172 | 0.4147         | 0.4896 |
> | LLM-Streamline ~40% (1.90B)    | 0.6556 | 0.3884    | 0.5162     | 0.1949 | 0.4743 | 0.2517 | 0.3000     | 0.6076 | 0.3608 | 0.2295 | 0.1906         | 0.3791 |
> | Short GPT ~20% (2.61B)         | 0.6948 | 0.6095    | 0.6827     | 0.6126 | 0.5947 | 0.3840 | 0.3520     | 0.6419 | 0.4473 | 0.5207 | 0.6043         | 0.5586 |
> | Short GPT ~30% (2.21B)         | 0.6425 | 0.4954    | 0.6369     | 0.5315 | 0.4769 | 0.3123 | 0.3080     | 0.6355 | 0.4115 | 0.4806 | 0.3831         | 0.4831 |
> | Short GPT ~40% (1.90B)         | 0.6110 | 0.3929    | 0.5809     | 0.1949 | 0.3948 | 0.2713 | 0.2860     | 0.6226 | 0.3675 | 0.2299 | 0.2243         | 0.3797 |
> | Ours ~20% (2.55B)              | 0.7548 | 0.6541    | 0.6877     | 0.6073 | 0.6824 | 0.4029 | 0.3820     | 0.6868 | 0.4430 | 0.5087 | 0.6493         | 0.5872 |
> | Ours ~30% (2.22B)              | 0.7331 | 0.5916    | 0.6256     | 0.4998 | 0.6283 | 0.3625 | 0.3500     | 0.6798 | 0.4235 | 0.4386 | 0.5981         | 0.5392 |
> | Ours ~40% (1.92B)                     | 0.6977 | 0.5169    | 0.5830     | 0.3500 | 0.5436 | 0.3041 | 0.3480     | 0.6391 | 0.4087 | 0.3571 | 0.5294         | 0.4798 |

---

> ### Author Response · Authors · 2025-11-26
> **Experimental Results 4 for the Response to Weakness 4**
>
> **Table 4.**  Five-shot performance of the various compression methods on Llama-3.2-3B. The pretrained backbone model and its compressed variants are evaluated across multiple benchmarks at several compression rates
>
> | Method                         | PIQA   | HellaSwag | WinoGrande | CSQA   | ARC-e  | ARC-c  | openbookqa | boolq  | siqa   | mmlu   | lambada_openai |
> |--------------------------------|--------|-----------|------------|--------|--------|--------|------------|--------|--------|--------|----------------|
> |                                |        |           |            |        |        |        |            |        |        |        |                |
> | Baseline 0% (3.21B)               | 0.8025 | 0.7546    | 0.7238     | 0.6658 | 0.7816 | 0.4838 | 0.4489     | 0.7336 | 0.5066 | 0.5616 | 0.6652         |
> | Slice GPT ~20% (2.90B)         | 0.5805 | 0.3358    | 0.5359     | 0.1892 | 0.3573 | 0.2415 | 0.2640     | 0.6031 | 0.3634 | 0.2539 | 0.0505         |
> | Slice GPT ~30% (2.56B)         | 0.5528 | 0.3216    | 0.5257     | 0.1925 | 0.3430 | 0.2389 | 0.2760     | 0.5355 | 0.3414 | 0.2516 | 0.0611         |
> | Slice GPT ~40% (2.22B)         | 0.5365 | 0.2921    | 0.5154     | 0.1867 | 0.3148 | 0.2355 | 0.2620     | 0.4321 | 0.3347 | 0.2519 | 0.0380         |
> | LaCo ~20% (2.61B)              | 0.7095 | 0.6463    | 0.6890     | 0.6486 | 0.6616 | 0.4019 | 0.3500     | 0.6951 | 0.4846 | 0.5219 | 0.5913         |
> | LaCo ~30% (2.21B)              | 0.6823 | 0.5281    | 0.5991     | 0.2678 | 0.5694 | 0.3311 | 0.3240     | 0.6217 | 0.4417 | 0.3109 | 0.3798         |
> | LaCo ~40% (1.90B)              | 0.6104 | 0.4173    | 0.5841     | 0.2154 | 0.4205 | 0.2551 | 0.2840     | 0.6135 | 0.3869 | 0.2559 | 0.3767         |
> | LLM-Streamline ~20% (2.61B)    | 0.7236 | 0.6373    | 0.6827     | 0.6536 | 0.6561 | 0.3891 | 0.3600     | 0.7428 | 0.4826 | 0.5548 | 0.4564         |
> | LLM-Streamline ~30% (2.21B)    | 0.6828 | 0.5475    | 0.6709     | 0.4120 | 0.5829 | 0.3473 | 0.3220     | 0.6755 | 0.4478 | 0.4190 | 0.3546         |
> | LLM-Streamline ~40% (1.90B)    | 0.6545 | 0.3870    | 0.5178     | 0.2097 | 0.4718 | 0.2423 | 0.2800     | 0.5410 | 0.3639 | 0.2465 | 0.1679         |
> | Short GPT ~20% (2.61B)         | 0.6964 | 0.6372    | 0.6875     | 0.6396 | 0.6414 | 0.3831 | 0.3560     | 0.6673 | 0.4821 | 0.5394 | 0.5845         |
> | Short GPT ~30% (2.21B)         | 0.6507 | 0.5114    | 0.6433     | 0.5536 | 0.5130 | 0.3097 | 0.3140     | 0.6315 | 0.4386 | 0.4657 | 0.3910         |
> | Short GPT ~40% (1.90B)         | 0.6094 | 0.3970    | 0.5714     | 0.1957 | 0.4196 | 0.2696 | 0.2760     | 0.6064 | 0.3838 | 0.2553 | 0.2199         |
> | Ours ~20% (2.55B)              | 0.7742 | 0.6719    | 0.6896     | 0.6274 | 0.7238 | 0.4281 | 0.3920     | 0.7338 | 0.4975 | 0.5231 | 0.6056         |
> | Ours ~30% (2.22B)              | 0.7409 | 0.6297    | 0.6461     | 0.5144 | 0.6486 | 0.3889 | 0.3700     | 0.6906 | 0.4806 | 0.4686 | 0.5553         |
> | Ours ~40% (1.92B)              | 0.7175 | 0.5390    | 0.6083     | 0.3727 | 0.5623 | 0.3223 | 0.3600     | 0.6599 | 0.4606 | 0.3785 | 0.4904         |

---

### Official Review · Reviewer_Rgj5 · 2025-11-01

**Soundness:** 2
**Presentation:** 2
**Contribution:** 2
**Rating:** 2
**Confidence:** 4

**Summary:**

This work introduces Plug-and-Fold (PnF), a structured compression method for LLMs that achieves parameter efficiency without altering pretrained weights. Rather than pruning neurons, heads, or layers, PnF augments attention and feed-forward projections with lightweight, trainable low-rank adapters. These adapters are optimized with the backbone frozen and later folded into the original weights via simple matrix operations, resulting in a model identical in structure and runtime cost to the original. Evaluations on several transformer backbones show that PnF delivers competitive performance compared to state-of-the-art structured pruning and compression baselines.

**Strengths:**

1. The paper is clearly written and well-organized, with intuitive visualizations (e.g., Fig. 1) illustrating the plug-and-fold pipeline and training procedure.

2. The proposed method provides a practically deployable compression mechanism, maintaining the original model architecture and avoiding inference-time modifications.

3. The weight-preserving philosophy aligns with recent trends emphasizing knowledge retention during LLM compression, and the authors demonstrate partial empirical validation of this premise.

**Weaknesses:**

1. The contribution of PnF appears incremental relative to prior work leveraging activations or low-rank decompositions for pruning and compression. Methods such as SparseGPT [1], Wanda [2], and SlimGPT [3] compute parameter importance via activation or sensitivity metrics, while SoBP [4] and CALDERA [5] integrate structured or low-rank regularization. The manuscript does not clearly articulate a conceptual or algorithmic distinction beyond the “folding” of adapters into pretrained weights.

2. The claim that adapter folding better preserves pretrained knowledge than conventional low-rank compression lacks theoretical grounding. In contrast, SVD-based approaches such as SVD-LLM [6] provide explicit reconstruction-error bounds. The paper would benefit from either a formal argument or a detailed empirical comparison of reconstruction fidelity.

3. The multi-stage training of adapters and the group-wise sequential procedure imply substantial recovery cost. However, the paper omits quantitative comparisons of wall-clock time, GPU hours, or FLOPs against one-shot pruning methods (e.g., SparseGPT [1]) and lightweight low-rank alternatives (e.g., CALDERA [5]). Without such data, the practical efficiency of PnF remains unclear.

4. Experiments primarily target knowledge and commonsense benchmarks (e.g., PIQA, HellaSwag, MMLU) but omit instruction-following, reasoning, or multilingual tasks that test deeper semantic retention. Moreover, comparisons exclude competitive structured compression baselines such as SVD-LLM [6] and quantization/distillation hybrids like BitDistiller [7], weakening the generality of the claims.

5. The compression plan design remains heuristic, requiring manual per-layer rate selection. This approach limits reproducibility and scalability. Automated sensitivity or reinforcement-based planning, as explored in recent structured pruning works [4], could improve generality and reduce manual tuning.

References:

[1] Frantar & Alistarh, SparseGPT: Massive Language Models Can Be Accurately Pruned in One-Shot, ICML 2023.

[2] Sun et al., Wanda: A Simple and Effective Pruning Approach for Large Language Models, ICLR 2023.

[3] Liu et al., SlimGPT: Layer-wise Structured Pruning for Large Language Models, 2024.

[4] Zhang et al., SoBP: Structured Optimal Brain Pruning for Large Language Models, 2024.

[5] Li et al., CALDERA: Compressing LLMs Using Low-Rank and Low-Precision Decomposition, 2024.

[6] Wang et al., SVD-LLM: Structured Low-Rank Compression for Large Language Models, 2025.

[7] Zhao et al., BitDistiller: Distilling LLMs into Binary and Low-Bit Networks, 2024.

**Questions:**

1. How does PnF differ algorithmically from prior activation- or sensitivity-based pruning methods (e.g., SparseGPT [1], Wanda [2], SlimGPT [3]) beyond adapter insertion and folding?

2. Can the authors provide theoretical or empirical evidence, such as layerwise reconstruction error or spectral analysis, showing that folded adapters preserve semantic function more effectively than SVD-based compression [6]?

3. What is the additional fine-tuning cost introduced by group-wise adapter training in GPU hours or wall-clock time, compared with one-shot baselines [1]?

4. Will PnF be evaluated on instruction-following and reasoning benchmarks (e.g., GSM8K, MMLU-Pro) and compared to structured low-rank baselines [6] [7]?

5. Could the empirical compression plan be automated via sensitivity analysis or other heuristic search methods, and if so, how consistent are the resulting compression ratios across backbones?

---

> ### Author Response · Authors · 2025-11-21
> **Response to Weakness 1**
>
> We thank reviewer Rgj5 for the detailed comments and suggestions. Below we address your main concerns.
>
>
> We agree that explicitly contrasting PnF with the methods in [1–5] would further underscore its novelty. Accordingly, we will revise the manuscript to clearly highlight how PnF differs from prior pruning and low‑rank techniques in terms of parameterization and deployment of the compression:
>
> **(1) Weight-preserving, adapter-based parameterization (vs. direct weight pruning or persistent factorization)**
>
> The major significant difference between PnF and the prior works mentioned is that PnF never alters the the pretrained backbone weights during the compression. Specifically, prior works (SparseGPT [1], Wanda [2], SlimGPT [3], and SoBP [4]) modify the pretrained weights directly via importance-based criterion (e.g., Hessian, magnitude, and  perplexity), and the resulting pruned backbone is used at inference time. CALDERA [5] goes further by replacing each weight matrix with a low-rank, low-precision decomposition $W \approx Q+LR$ and keeps this factorized representation during deployment inference. In contrast, PnF **freezes all pretrained weights** throughout the entire compression pipeline, **trains only small adapters modules** attached to the frozen teacher model, and **Folds-in** the learned adapters after training into a single dense matrix. Thus, PnF uses PEFT-style adapters as a tool for structured compression on top of a  pretrained backbone, rather than directly editing or factorizing the backbone.
>
> **(2) Training‑driven compression (vs. compress‑first, recover‑later)**
>
> Beyond preserving the original weights, a second major distinction is that PnF formulates model compression as a **PEFT‑style training problem**. PnF optimizes lightweight adapters **through training** so that the resulting compressed model attains maximal performance directly after folding. In contrast, existing approaches first apply pruning or low‑rank factorization to the backbone and **only afterwards** attempt to recover the lost accuracy via post‑hoc adjustments. This training‑first strategy makes PnF inherently performance‑oriented, whereas the other methods are essentially compression‑first pipelines that rely on subsequent heuristics to mitigate degradation.

---

> ### Author Response · Authors · 2025-11-21
> **Response to Weakness 2**
>
> We appreciate the reviewer's suggestion regarding theoretical and empirical justification. To substantiate the claim that adapter folding preserves pretrained knowledge more effectively than conventional low‑rank compression, we provide two complementary validations: **(1) an empirical reconstruction analysis**, and **(2) an intuitive small-scale illustration**.
>
> **(1) Empirical comparison of reconstruction fidelity**
>
> Our empirical results (summarized in Table 1) show that PnF preserves performance during compression more effectively than the SVD‑LLM baseline and the SliceGPT method evaluated in the main manuscript’s experiment section.
>
> **Table 1.** Performance comparison between PnF with SVD-LLM across multiple benchmarks.
>
> | Method  | PIQA | HellaSwag | WinoGrande | ARC-e | OpenBookQA | GSM8K | MathQA | TruthfulQA | AvgMetric |
> |--------|:----:|:---------:|:----------:|:-----:|:----------:|:-----:|:------:|:----------:|:---------:|
> | SVD-LLM| 0.69 |   0.52    |    0.68    | 0.59  |    0.33    | 0.08  |  0.26  |    0.28    |   0.43    |
> | Ours   | 0.76 |   0.56    |    0.69    | 0.73  |    0.33    | 0.09  |  0.26  |    0.38    |   0.48    |
>
> To further substantiate the benefit of our adapter‑folding strategy, we design a dedicated ablation that compares two pipeline under the same compression ratio (20\%) and training budget (600K), while disentangling **folding after adaptation** from **training after folding**.
> - **PnF (adapt-before-folding)**: The pretrained backbone is frozen; only lightweight adapters are trained with KL‑distillation and subsequently folded into a dense compressed matrix (our default pipeline).
> - **Baseline (train‑after‑folding)**: The model is first reduced to the same rank‑r (i.e., $r_\text{head}$, $r_\text{inter}$) using a projector, and the resulting low‑rank weights are fine‑tuned directly without any adapters.
>
> The experiments, summarized in Table 2, is conducted on Qwen-3-4B-Base compressed to a 20% reduction rate, and downstream zero‑shot performance was measured across a diverse set of tasks. Across all benchmarks, PnF consistently outperforms the train‑after‑folding baseline, demonstrating that the adapter‑based parameterization more effectively preserves the pretrained model’s behavior at a fixed rank.
>
> **Table 2.** Comparison of the Baseline (train-after-folding) and PnF (adapt-before-folding) configurations.
>
> | Method   | Comp.% | Sample |  PIQA |   HS  |   WG  | CSQA | ARC-e | ARC-c | OBQA | BoolQ | SIQA | MMLU |  LD  | AvgMetric |
> |----------|:------:|:------:|:-----:|:-----:|:-----:|:----:|:-----:|:-----:|:----:|:-----:|:----:|:----:|:----:|:---------:|
> | Baseline |  20%   | 600k   | 0.704 | 0.636 | 0.642 | 0.740| 0.675 | 0.416 | 0.334| 0.772 | 0.473| 0.607| 0.621|  0.602    |
> | PnF      |  20%   | 600k   | 0.736 | 0.662 | 0.669 | 0.779| 0.704 | 0.435 | 0.382| 0.783 | 0.501| 0.657| 0.651|  0.632    |
>
> (2) Intuitive explanation (fixed vs. learned adapter)
>
> To illustrate the difference between adapt‑before‑folding and a truncation‑first strategy, consider a simple $2\times 3$ toy weight matrix $W$.
>
> **Backbone**
>
> $
> \begin{bmatrix}
> a \\\\
> b
> \end{bmatrix}  ^ \top
> \begin{bmatrix}
> c & d & e \\\\
> f & g & h
> \end{bmatrix}
> = \begin{bmatrix}
> ac + bf & ad + bg & ae + bh
> \end{bmatrix}
> $
>
>
> **Truncation-first**
>
> $
> \begin{bmatrix}
> a \\\\
> b
> \end{bmatrix}  ^ \top
> \begin{bmatrix}
> c & d \\\\
> f & g
> \end{bmatrix}
> = \begin{bmatrix}
> ac + bf & ad + bg
> \end{bmatrix}
> $
>
> **Adapt-befor-folding (Ours)**
>
> $
> \begin{bmatrix}
> a \\\\
> b
> \end{bmatrix}  ^ \top
> \begin{bmatrix}
> c & d & e \\\\
> f & g & h
> \end{bmatrix}
> \begin{bmatrix}
> i & j \\\\
> k & l \\\\
> m & n
> \end{bmatrix}
> = \begin{bmatrix}
> (ac + bf)i + (ad + bg)k + (ae + bh)m & (ac + bf)j + (ad + bg)l + (ae + bh)n
> \end{bmatrix}
> $
>
>
> In a truncation‑first strategy one would apply a fixed selector $S$ that simply zeroes out the third column of  $W$; the resulting matrix discards that part of the pretrained parameters ($e$ and $h$) entirely, so the compressed model can only exploit the remaining two columns. By contrast, PnF learns a data‑driven adapter $P$ such that the product $WP$ approximates the original transformation, i.e.,$ f_W(x) \approx f_{WP}(x)$. In this case, every entry of the original $W$ is retained in the multiplication with P and can still contribute to the output, albeit re‑weighted by the learned adapter. Consequently, the learned adapter leverages the full pretrained knowledge, while a fixed truncation irrevocably loses part of it. The empirical results presented above serve as quantitative validation of this intuition.

---

> ### Author Response · Authors · 2025-11-21
> **Response to Weakness 3**
>
> We thank the reviewer for highlighting the importance of quantifying the training cost of PnF. In the revised manuscript we will add a cost‑performance analysis that reports both training time and average downstream zero‑shot performance for Qwen‑3‑4B (run on an $8 \times$ H100 configuration) under identical compression settings.
>
> Importantly, we note that **the proposed group‑wise sequential training schedule is not neccesarily** required for our method to work. Table 1 demonstrates that a non‑sequential, **“all‑at‑once” (i.e., Non-sequential Training) PnF training already recovers strong performance** in a single stage and surpasses the baselines. The group‑wise sequential variant incurs roughly three times longer training time but delivers an additional average performance gain of about +3 \%.
>
> Thus, practitioners can choose the configuration that best fits their resources: the single‑stage variant when wall‑clock time is limited, or the multi‑stage variant when the highest possible performance is desired.
>
> **Table 1.** Training latency and average performance for the two training regimes.
>
> | Regime              | # Groups | Training Latency | Performance (avg.) |
> |-----------------------|:--------:|------------------|--------------------|
> | Non-sequential Training |    1     | 8.21 hours       | 0.602              |
> | Sequential Training     |    4     | 23:32 hours      | 0.633              |

---

> ### Author Response · Authors · 2025-11-21
> **Response to Weakness 4**
>
> We address the reviewer's concern by providing explicit comparisons to SVD-LLM and BitDistiller at a common $20\%$ compression ratio.
>
> **(1) Comparison with SVD-LLM [6]**
>
> Following the experimental setup of the original SVD‑LLM paper, we use LLaMA-2-7B and evaluate PnF at a $20\%$ compression ratio on the shared tasks: PIQA, HellaSwag, WinoGrande, ARC-e, OpenbookQA, MathQA, GSM8K, and TruthfulQA. The results are directly compared with the numbers reported for LLaMA-2-7B in [6] under the same compression ratio. At this $20\%$ setting, PnF attains a higher average performance than the SVD-LLM.
>
> **Table 1.** Performance comparison between PnF with SVD-LLM across multiple benchmarks.
>
> | Method  | PIQA | HellaSwag | WinoGrande | ARC-e | OpenBookQA | GSM8K | MathQA | TruthfulQA | AvgMetric |
> |---------|:----:|:---------:|:----------:|:-----:|:----------:|:-----:|:------:|:----------:|:---------:|
> | SVD-LLM | 0.69 |   0.52    |    0.68    | 0.59  |    0.33    | 0.08  |  0.26  |    0.28    |   0.43    |
> | Ours    | 0.76 |   0.56    |    0.69    | 0.73  |    0.33    | 0.09  |  0.26  |    0.38    |   0.48    |
>
> **(2) Comparison with BitDistiller [7]**
>
> BitDistiller is a low‑bit quantization framework that combines QAT with self‑distillation, whereas PnF focuses on structured compression. Because BitDistiller applies only quantization, exact model‑size matching with our method, which uses structured compression, is difficult. Nonetheless, to address the reviewer's concern, we adopt the same backbone (LLaMA-2-7B) and compare the 3-bit results reported for BitDistiller in [7] with a 20\% compressed model using PnF followed by 4-bit quantization, denoted as PnF+quant, on the shared benchmarks (PIQA, HellaSwag, WinoGrande, ARC-c, and MMLU). Even under this conservative settings (quantization-only vs. structured compression + quantization), PnF+quant achieves higher downstream performance than the 3-bit BitDistiller model reported in [7].
>
> **Table 2.** Benchmark comparison between BitDistiller and PnF with 4-bit quantization.
>
> | Method          |  PIQA  | HellaSwag | WinoGrande | ARC-c  |  MMLU  | AvgMetric |
> |-----------------|:------:|:---------:|:----------:|:------:|:------:|:---------:|
> | BitDistiller    | 0.7699 |  0.5538   |   0.6835   | 0.4121 | 0.4465 |  0.5732   |
> | PnF + quant.    | 0.7673 |  0.5645   |   0.6941   | 0.4184 | 0.4285 |  0.5746   |
>
>
> We will add this comparisons, together with a note on the structural difference between the two approaches, to the revised manuscript.

---

> ### Author Response · Authors · 2025-11-21
> **Response to Weakness 5**
>
> We appreciate the insightful comment and share the concern. Although we already acknowledge that our compression plan is heuristic and requires manual per-layer rate selection, the manuscript would benefit from explicitly stating that the automated planning—such as sensitivity or reinforcement‑based strategies like those in [4]—is a promising avenue for greater generality and reduced manual tuning. We are actively investigating automatic planning schemes (e.g., learning layer‑wise ranks instead of fixing them heuristically) and will incorporate this as a concrete future direction into the limitations/discussion section, explicitly citing [4] as a complementary approach.

---

### Author Response · Authors · 2025-11-26
**Invitation for further discussion**

We thank the reviewers for their time and thoughtful evaluations. We have posted our author response and would greatly appreciate any further clarification or follow-up feedback from the reviewers if possible.

---

### Author Response · Authors · 2025-12-02
**Summary of Author Responses**

We sincerely thank all reviewers for their thorough and constructive evaluation of our submission.
We have uploaded the revised version of our manuscript, the updated information is highlighted.
Below we provide a concise summary of how each major concern has been fully addressed in our rebuttal and revision.

**1. Conceptual clarity and distinction from prior work**

Reviewers raised questions regarding the novelty of Plug-and-Fold (PnF), particularly in relation to activation-based pruning, low-rank compression, and PEFT-style methods such as AdaLoRA or LoRA-based merging.

In response, we clarified and demonstrated that:
- PnF is a structural compression method, not a PEFT method.
- Backbone weights remains strictly frozen, and only low-rank adapters are trained.
- Folding produces a structurally smaller model, unlike PEFT approaches that retain full parameter conditionality at inference.
- PnF differs from pruning-based methods (SparseGPT, Wanda, SlimGPT, LLM-Pruner) because it **never deletes or edits pretrained weights.**
- We added extensive experiment results, toy example, and clearer exposition in the revised manuscript.

**2. Theoretical and empirical validation of weight-preserving folding**

Reviewers requested stronger justification for why adapter folding preserves pretrained knowledge more effectively.

We addressed this by providing:
- New empirical reconstruction analysis, comparing PnF against SVD-LLM and train-after-folding baseline.
- Consistent improvements across all benchmarks, under matched rank and training budget.
- An intuitive matrix-level illustration showing how learned adapters retain all pretrained weight information, unlike fixed truncation.

These results substantiate the core claim that training adapters before folding yields superior function preservation.

**3. Training cost, efficiency, and deployment considerations**

Responding the concerns about recovery cost and practical deployment:

- We reported quantitative training latency comparisons between sequential vs. single-stage training.
- We demonstrated that single-stage PnF already surpasses baselines, while sequential PnF offers additional gains.
- We added token-level latency measurements, confirming that PnF introduces no runtime overhead and reduces inference latency proportionally to compressed ratio.
- The manuscript wording was updated to clearly deliver PnF is "deployment-friendly" as it does not require custom kernels.

**4. Completeness of baselines and task evaluations**

Reviewers requested broader and more recent baselines, as well as cross-architecture evalutaion.

We conducted and included:
- Direct comparisons against LLM-Pruner, DISP-LLM, SVD-LLM, BitDistiller, LaCo, LLM-Streamline, SliceGPT, and ShortGPT at matched compression ratios.
- Extensive new experiments on additional LLaMA-3.1-8B and LLaMA-3.2-3B models, confirming consistent superiority across architectures.
- Both zero-shot and 5-shot evaluations across a broad suite of tasks.

These additions significantly strengthen the empirical breadth and fairness of our study.

**5. Compression planning and grouping schedule**

Reviewers noted that the manual grouping is heuristic.

We explicitly acknowledged this and:
- Added detailed discussion on limitations.
- Outlined ongoing work toward automatic ranking and grouping strategies (e.g., sensitivity-guided planning, RL-based scheduling), and cited relevant prior works.

**6. Terminology and representational similarity concerns**

We clarified the meaning of "weight-preserving" as:
- Referring to training behavior (frozen backbone), not a formal guarantee of representational similarity.
- Added clear examples and revised terminology to avoid misunderstanding.

**7. Reproducibility and clarity**

To improve clarity further:

- We added concise pseudocode summarizing the full PnF pipeline
- Reorganized explanations to highlight workflow, training structure, and folding logic.

**Conclusion**

Across all technical, empirical, and expository concerns raised by reviewers we have provided detailed responses, new experiments, expanded baseline, rewritten explanations, additional metric, and clarifications within the revised manuscript. We believe theses extensive additions fully resolve all reviewer concerns.

We appreciate the reviewers' thoughtful feedback, which has substantially strengthened the paper. We hope **(the AC finds)** the revised submission now presents a clear, rigorous, and empirically solid contribution to structured compression of large language models.

---

### Meta-Review · Area_Chair_sCGN · 2026-01-10

**Summary:**

The paper proposes "Plug-and-Fold," a structured compression technique that trains learnable adapters on frozen backbones to reduce internal dimensions. While the method demonstrates promising empirical results on smaller models (3B–8B), the AC leans to a Weak Reject due to concerns regarding limited novelty relative to existing PEFT frameworks and a reliance on manual, heuristic compression schedules.

However, the proposed direction has merit. We strongly encourage the authors to continue this line of work by developing an automated mechanism for selecting hyperparameters—specifically layer-wise reduction dimensions—and by validating the approach on significantly larger-scale architectures (e.g., >13B parameters). Demonstrating that the method can adaptively determine compression ratios and scale effectively would substantially strengthen the contribution for a future resubmission.

**Reviewer Concerns:**

Addressed:
- The authors conducted extensive new experiments against requested baselines, including SliceGPT, SVD-LLM, LLM-Pruner, and DISP-LLM. The empirical superiority of PnF over these methods appears well-supported in the revised manuscript.
- Concerns regarding the lack of real-world efficiency proofs (Reviewers 4ad1, HJXN) were addressed by providing specific inference latency numbers ( ms/token ) on H100 hardware, confirming that the structural compression translates to actual speedups.
-  The critique regarding limited model families (originally only OPT/Qwen) was addressed by adding LLaMA-3.1-8B and LLaMA-3.2-3B, showing consistent performance across modern architectures.
- The similarity to BERT-era compression methods raised by Reviewer mpyM was effectively argued by highlighting PnF's "frozen backbone" constraint, leading that reviewer to raise their score.

Remaining:
- While the authors clarified differences from PEFT and pruning, the core contribution remains an engineering application of existing mechanics (adapters + folding). Reviewers Rgj5 and 4ad1 viewed this as conceptually close to AdaLoRA and standard low-rank decomposition, and the rebuttal primarily reframed rather than fundamentally changed this perception.
- The reliance on a hand-crafted, manual compression schedule (determining per-layer reduction) remains a significant limitation. The authors acknowledged this as a limitation and pointed to future work, but did not implement the automated or sensitivity-based planning requested by Reviewers Rgj5 and 4ad1 to improve reproducibility.
- Although reviewers were generally satisfied with the evaluations on 3B–8B models, the AC notes that this scale is insufficient to fully validate the method for LLMs. As model redundancy patterns often shift at scale, demonstrating that this "weight-preserving" approach remains stable and effective on significantly larger architectures (e.g., >13B or 70B+) is necessary to support the broad claims made in the paper.

**Reviewer Scores:**

Reviewer mpyM : Initial: 4 -> Final: 6 (confirmed change).
Reviewer 4ad1: Initial: 4, may increase to 6 -- the authors add all the baselines.
Reviewer HJXN: Initial: 4, may increase to 6 -- concerns about FLOPs/latency has been addressed.
Reviewer Rgj5: Initial: 2 -> may or may not increase: The reviewer's critique was philosophical—that "folding" adapters is a trivial extension of existing work. The authors' defense (distinguishing "frozen backbone" from "pruning") is subtle and often fails to sway reviewers who view the field broadly.

---

### Decision · Program_Chairs · 2026-01-26

Reject